# CHAINMPQ: INTERLEAVED TEXT-IMAGE REASONING CHAINS FOR MITIGATING RELATION HALLUCINATIONS

**Yike Wu**[1]    **Yiwei Wang**[2*]    **Yujun Cai**[1,3]
[1]University of Queensland    [2]University of California, Merced    [3]Ant Group
yikewu66@gmail.com; yiweiwang2@ucmerced.edu
https://yike-wu.github.io/chainmpq-project

## ABSTRACT

While Large Vision-Language Models (LVLMs) achieve strong performance in multimodal tasks, hallucinations continue to affect their reliability. Among the three categories of hallucinations, which include object, attribute, and relation, relation hallucinations account for the largest proportion but have received the least attention. To address this challenge, we propose **ChainMPQ** (**M**ulti-**P**erspective **Q**uestions guided Interleaved Text-image Reasoning **Chain**), a training-free method that improves relational inference in LVLMs by utilizing accumulated textual and visual memories. ChainMPQ first extracts subject and object keywords from the question to enhance the corresponding image regions. It then constructs multi-perspective questions that focus on the three core components of a relationship: the subject, the object, and the relation that links them. These questions are sequentially input to the model, with textual and visual memories from earlier steps providing supporting context for subsequent ones, thereby forming an interleaved chain of image and text that guides progressive relational reasoning. Experiments on multiple LVLMs and benchmarks show that Chain-MPQ substantially reduces relation hallucinations, while ablation studies further validate the effectiveness of its three core modules.

## 1 INTRODUCTION

Large Language Models (LLMs) have made substantial advances in language understanding, generation, and reasoning Touvron et al. (2023b;a); Bai et al. (2023a). Extending these capabilities with visual inputs, Large Vision-Language Models (LVLMs) achieve strong performance on a range of multimodal tasks, including image captioning and visual question answering Ye et al. (2024); Li et al. (2023a); Bai et al. (2023b); Li et al. (2023b); Dai et al. (2023); Liu et al. (2024b). Nevertheless, LVLMs still exhibit hallucinations, producing outputs that contradict or overlook the visual evidence. Hallucinations in LVLMs can be broadly categorized into object, attribute, and relation hallucinations Bai et al. (2024). Object hallucinations refer to failures in recognizing entities, while attribute hallucinations involve misidentifying properties such as color or shape. Relation hallucination occurs when models correctly recognize objects but fail to infer the relationship between them, as shown in Figure 1.

Although prior work has substantially reduced object and attribute hallucinations through preference optimization Wu et al. (2025a); Yang et al. (2025), contrastive decoding Wu et al. (2025c); Zhang et al. (2025), and intermediate-layer modification Jiang et al. (2024; 2025); Wang et al. (2024), relation hallucination, which accounts for nearly 40% of all hallucinations Zheng et al. (2024), remains a salient challenge. Existing efforts to mitigate relation hallucinations have made some progress: dataset-driven approaches Xie et al. (2025) focus on constructing high-quality training and fine-tuning data, prompt-engineering methods Wu et al. (2025a) employ constrained perception prompts, Detect-then-Calibrate Zheng et al. (2024) selectively adjust the final logits with intermediate layer, and Triplet Description Wu et al. (2025b) converts the image into triplets, allowing the model to answer questions based on these structured descriptions instead of the raw image.

---

*Corresponding author.

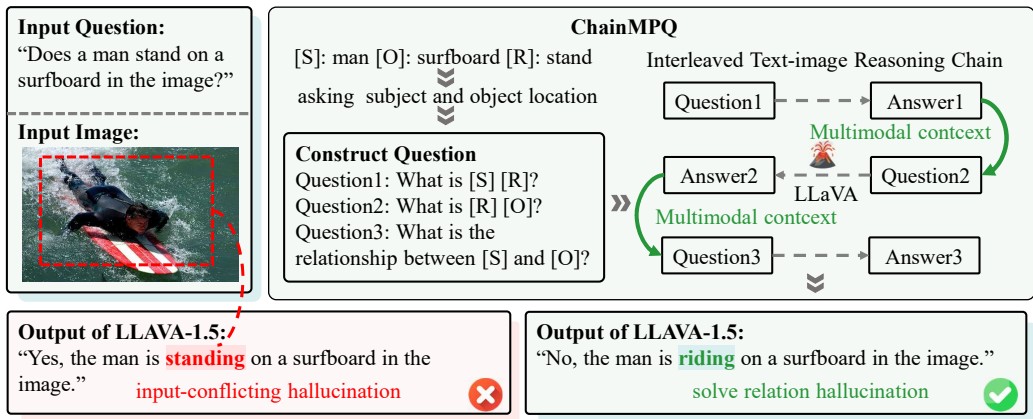

Figure 1: This figure illustrates a case in which an LVLM exhibits a relation hallucination, misrecognizing the true relation "riding" as "standing". With ChainMPQ, the model is guided to reason step by step and infer the correct relation.

Despite their promising results, existing methods targted at relation hallucinations treat relational reasoning as single-step inference, expecting models to simultaneously identify entities and determine relationships. This approach is likely to produce errors because it relies heavily on language priors rather than systematic visual analysis. Human relational reasoning, by contrast, follows a more structured process: first locating and identifying relevant objects, then examining their interactions, and finally synthesizing visual evidence to draw conclusions about relationships. This multi-step approach allows for more careful consideration of visual information and reduces the likelihood of inference errors.

Inspired by human reasoning process and drawing insights from Interleaved Modal Chain-of-Thought (ICoT) Gao et al. (2025), we propose **ChainMPQ** (**M**ulti-**P**erspective **Q**uestions guided Interleaved Text-image Reasoning **Chain**), a training-free framework that decomposes relational inference into manageable steps while maintaining relevant reasoning through accumulated multimodal memory. Our approach addresses the core challenge of relation hallucination by combining systematic question decomposition with progressive multimodal reasoning.

Specifically, ChainMPQ first extracts subject and object keywords from the question and uses them to drive cross-attention. Next, relations are decomposed into three components, and multiperspective questions are constructed by masking one component at a time. Finally, the interleaved chain is built by sequentially feeding the constructed questions into the model, using previous answers as textual context and earlier relevant visual tokens to adjust later attention maps. This process enables the model to reason over prior textual and visual memories and progressively resolve the relation. We demonstrate the effectiveness of ChainMPQ on four advanced and widely used models, where it consistently reduces relation hallucinations on relation-focused benchmarks. Our contributions are summarized as follows:

1. We introduce a subject–object–relation decomposition to generate multi-perspective questions, encouraging the model to focus on each core element of a relationship.
2. We design an interleaved chain mechanism that transfers textual and visual memories by using answers and attention maps from earlier steps to refine subsequent reasoning, thereby enabling progressive relational inference.
3. We apply ChainMPQ to multiple LVLMs and relation-focused benchmarks, demonstrating consistent reductions in relation hallucinations.

## 2 RELATED WORK

### 2.1 LARGE VISION-LANGUAGE MODELS (LVLMS)

Large Vision-Language Models (LVLMs) enhance the capabilities of traditional Large Language Models (LLMs) by incorporating visual inputs, thereby enabling them to perform complex multi-

modal tasks such as image captioning and visual question answering. Prominent LVLMs, including InstructBLIP Dai et al. (2023) and LLaVA Liu et al. (2024b), integrate pre-trained vision encoders with language models, effectively bridging the gap between the visual and textual modalities.

While these models have achieved remarkable progress in visual-language understanding Lai et al. (2024); Laurençon et al. (2024); Li & Li (2025); Peng et al. (2024), they continue to face challenges of hallucinations, particularly in the recognition of relationships Zheng et al. (2024); Nie et al. (2024) between objects. Our work extends these models by specifically addressing the problem of relation hallucinations, employing a step-by-step reasoning process guided by both textual and visual information.

## 2.2 RELATION HALLUCINATION IN LVLMS

Despite significant advancements in mitigating object and attribute hallucinations, relation hallucination remains a critical and underexplored issue Liu et al. (2024a); Wu et al. (2024). These hallucinations occur when a model accurately detects objects but fails to correctly identify the relationships between them, leading to misleading or incorrect responses in tasks such as Visual Question Answering (VQA) Shahgir et al. (2024). The significance of addressing relation hallucinations is highlighted by their prevalence and studies indicate that nearly 40% of hallucinations in LVLMs are related to relations between objects Zheng et al. (2024), yet this issue has not received as much attention as object or attribute hallucinations.

Recent research has proposed several evaluation benchmarks aimed at assessing the handling of inter-object relations. For example, MMRel Nie et al. (2024) features over 10k question-answer pairs across multiple domains, specifically designed to evaluate spatial, action, and comparative relations. Additionally, R-Bench Wu et al. (2024) includes a diverse set of questions that balance perception and cognition tasks, exposing limitations in the relation reasoning capabilities of contemporary models. Tri-HE Wu et al. (2025b) further extends evaluation by using triplets to represent knowledge, which allows it to assess both object and relation hallucinations at the same time. These benchmarks are valuable for systematically evaluating and enhancing the management of relation hallucinations. For considerations of dataset size and usage frequency, we evaluate our method using MMRel and R-Bench, which are larger and more widely used.

## 2.3 INTERLEAVED MODAL CHAIN-OF-THOUGHT (ICOT)

Interleaved Modal Chain-of-Thought (ICoT) Gao et al. (2025) extends the Chain-of-Thought (CoT) reasoning paradigm Wei et al. (2022) to multimodal tasks by integrating visual and textual reasoning steps. While traditional CoT methods, which primarily focus on text, struggle to handle dynamic visual state transitions, ICoT addresses this limitation by progressively updating intermediate visual states throughout the reasoning process. This enables the model to maintain coherent and grounded reasoning across both modalities. The dynamic interaction between visual and textual information in ICoT closely mirrors human cognitive processes, where visual and textual elements evolve concurrently to support decision-making. Recent advances in multimodal reasoning, including approaches such as Uni-CoT Qin et al. (2025) and visual reasoning frameworks Lin et al. (2025), emphasizes the importance of integrating visual feedback into reasoning. ICoT offers a flexible, scalable, and efficient framework for multimodal reasoning, establishing a new basic method for future research in this field. Given the promising potential of ICoT, we leverage its principles to build our own methods.

## 3 METHODS

### 3.1 PROBLEM FORMULATION

We address relation hallucination in visual question answering, where models correctly identify entities but fail to infer their relationships. Given an image $I$ and a relational question $Q$, the task is to generate an accurate Yes/No answer $A$. Relation hallucination occurs when a model detects both subject and object entities but provides incorrect relational judgments.

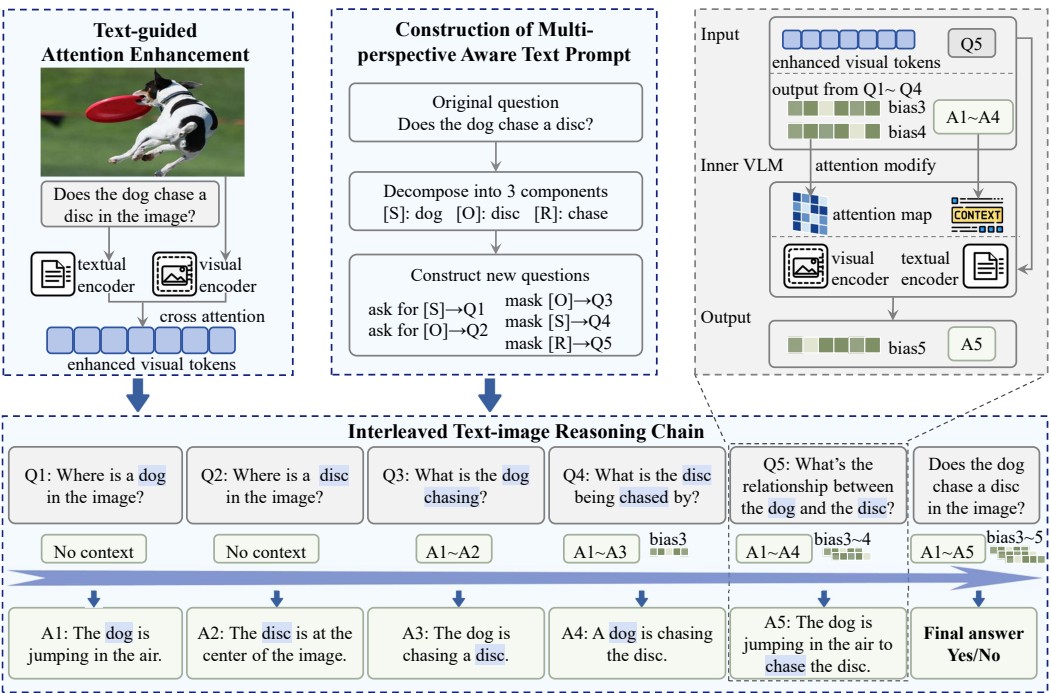

Figure 2: Overview of our proposed ChainMPQ. It comprises three modules. Text-guided Attention Enhancement: extracts subject, object, and relation, using cross-attention to emphasize relation-relevant visual regions; Multi-Perspective Aware Text Prompt: constructs five new questions based on these elements from different perspectives; Interleaved Text-image Reasoning Chain: sequentially inputs the questions, using each answer $A_i$ and its top-K active visual tokens to form mask $M_i$ as a bias when calculating subsequent attention maps. The original question is then answered to produce the final output and evaluation metrics.

## 3.2 OVERVIEW

ChainMPQ is a training-free framework designed to mitigate relation hallucinations through progressive multimodal reasoning. Unlike existing approaches that rely solely on textual prompting, our method leverages both textual answers and visual attention patterns across reasoning steps. For procedural details of the algorithm, see Algorithm 1.

The framework operates in three stages: (1) enhancing visual representations of entities mentioned in the question through text-guided attention mechanisms, (2) decomposing the original relational query into multiple complementary questions that target individual components of the relationship, and (3) constructing an interleaved chain where accumulated textual and visual information guides subsequent reasoning. This progressive approach enables the model to systematically analyze relationships rather than making immediate judgments based on surface patterns.

## 3.3 TEXT-GUIDED ATTENTION ENHANCEMENT

Accurate relational reasoning requires precise entity localization. We extract subject and object keywords from the input question using the spaCy NLP toolkit. These keywords are encoded to obtain representations $X \in \mathbb{R}^{N \times d_t}$, where $N$ denotes the number of keywords (typically two, corresponding to the number of subject and object), and $d_t$ represents the text feature dimension.

The input image is processed by the visual encoder to obtain visual features $V \in \mathbb{R}^{M \times d_v}$, where $M$ is the number of image patches and $d_v$ is the dimension of the visual features. We apply cross-attention to enhance image regions corresponding to the extracted keywords, where the image features $V$ serves as the Query vector and the keyword text $X$ as both the Key and Value vectors:

$$V' = \text{softmax}\left(\frac{VX^{\mathrm{T}}}{\sqrt{d_t}}\right) X \tag{1}$$

---

**Algorithm 1:** ChainMPQ

---

**Input:** Image $I$, relational question $Q$
**Output:** Final answer $A$
**Hyperparams:** $\lambda$, $k_{\max}$
**Symbols:** Textual context $\mathcal{T}$, visual memory $\mathcal{V}$
**Module 1: Enhance visual tokens**
    Extract keywords $K$ from $Q$;
    Encode $V \leftarrow \mathrm{VisualEncoder}(I)$ and $X \leftarrow \mathrm{TextEncoder}(K)$;
    Obtain enhanced visual tokens $V' \leftarrow \mathrm{CrossAttn}(X, V)$.
**Module 2: Construct sub-questions**
    Decompose $Q$ into $[S], [O], [R]$;
    Generate complementary sub-questions $Q_{1..5}$.
**Module 3: Interleaved text–image reasoning chain**
    Q1 and Q2: localize $[S]$ and $[O]$ using $V'$, update $\mathcal{T}$;
    Q3 to Q5: ask relation-focused questions and build top-$k$ mask $M_i$ with $k \leq k_{\max}$;
    Apply bias scaled by $\lambda$, update $\mathcal{T}$ and $\mathcal{V}$;
    Answer original question: decode with $V'$, $\mathcal{T}$ and $\mathcal{V}$ to produce $A$.

---

This operation produces enhanced visual tokens $V'$ that emphasize subject and object regions, establishing a foundation for accurate relational inference in subsequent steps.

## 3.4 Construction of Multi-Perspective Aware Text Prompt

We decompose the original question into five complementary questions that examine different aspects of the relationship. Specifically, the original input question typically contains three components: the subject, the object, and the relation between them. The first two questions focus on entity localization by asking about the location of the subject and object respectively. The remaining three questions employ a masking strategy where one component is masked while the other two are used to construct new queries.

To be specific, we mask the object to generate a question about what the subject is interacting with, mask the subject to ask what the object is being affected by, and mask the relation to inquire about the general relationship between the entities. For example, "Does the dog chase a disc in the image?" becomes five questions: entity localization ("Where is the dog/disc?") and three relation-focused queries generated through systematic masking, as shown in Figure 2.

This decomposition strategy encourages the model to analyze individual components before making final relational judgments, reducing reliance on language priors and improving systematic reasoning.

## 3.5 Interleaved Text-Image Reasoning Chain

Unlike previous text-only prompting methods Wu et al. (2025a), we sequentially process the enhanced visual tokens from Section 3.3 and constructed questions from Section 3.4 using an interleaved chain that transfers both textual and visual information across reasoning steps. For each question $Q_i$, we encode it using the text encoder to obtain question tokens, while maintaining a context $C_i$ that accumulates answers from previous steps.

The model generates answer $A_i$ for question $Q_i$ using the enhanced visual tokens $V'$, current context $C_i$, and any accumulated visual memory from earlier steps. The first and second questions are answered directly without adding any contextual information. Starting from the third question, we extract attention weights associated with the keyword tokens in each question from the final $n$ decoder layers to capture the model's focus on visual regions:

$$\mathrm{Attn}_i = \frac{1}{|T| \cdot n} \sum_{t \in T} \sum_{\ell = L-n}^{L-1} \mathrm{Attn}^{(\ell)}[t, :] \tag{2}$$

where T contains indices of keyword tokens, L is the total number of decoder layers, and $\mathrm{Attn}^{(\ell)}$ represents the attention matrix at layer $\ell$.

We select the top-k visual tokens with highest attention scores using an entropy-based adaptive strategy that adjusts $k$ based on attention concentration:

$$k = \left| k_{\max} \cdot \hat{H}(\text{Attn}_i) \right| \qquad I_{\text{topk},i} = \text{TopK}(\text{Attn}_i, k) \qquad (3)$$

where $\hat{H}(\text{Attn}_i)$ represents the normalized entropy of the attention distribution, $I_{\text{topk},i}$ denotes the set of visual tokens most attended to for the $i$-th question and $k_{max}$ is set to 20 in our experiments.

The selected tokens form a bias mask $M_i$ that guides attention in subsequent steps:

$$M_{i,j} = \begin{cases} \frac{\text{Attn}_{i,j}}{\sum_{j \in I_{\text{topk},i}} \text{Attn}_{i,j}} & j \in I_{\text{topk},i} \\ 0 & \text{othercases} \end{cases} \qquad (4)$$

where entries corresponding to $I_{\text{topk},i}$ are assigned normalized attention scores and all others are set to zero, and $M_{i,j}$ denoting the $j$-th element in $M_i$.

For subsequent questions, the attention mechanism incorporates this visual bias:

$$\alpha_i = \lambda \cdot Conf_{\text{prev}_i} \qquad \text{Attn}_{i+1} = \text{softmax}\left( \frac{QK^\top}{\sqrt{d_k}} + \alpha_i \cdot M_i \right) V \qquad (5)$$

where $\alpha_i$ is a confidence-based weight that increases with answer certainty.

For multi-round previous visual bias, we calculate a weighted average:

$$\text{Attn}_{i+1} = \text{softmax}\left( \frac{QK^\top}{\sqrt{d_k}} + \frac{1}{\sum_{j=3}^{i} \alpha_j} \sum_{j=3}^{i} \alpha_j \cdot M_j \right) V \qquad (6)$$

This mechanism enables the model to maintain visual focus across reasoning steps while progressively building understanding of the relationship. The accumulated multimodal evidence culminates in a final answer that reflects systematic relational analysis rather than surface-level pattern matching.

# 4 EXPERIMENTS

## 4.1 EXPERIMENTAL SETUP

**Evaluated LVLMs.** We evaluate our method on four open-source LVLMs: LLaVA-1.5-7B Liu et al. (2024b), InstructBLIP-7B Dai et al. (2023), Qwen2.5-VL-7B Bai et al. (2025), and InternVL3.5-8B Wang et al. (2025). The first two models are widely used and represent classic LVLM designs. The last two models are more recent and show stronger performance in many multimodal tasks. LLaVA-1.5 combines a CLIP visual encoder with Vicuna-7B for multimodal reasoning, while InstructBLIP-7B follows the BLIP-2 design, linking a frozen vision encoder to a language model through a Q-Former. Qwen2.5-VL is built on the Qwen2.5 language model and uses a redesigned ViT architecture to provide strong visual understanding. InternVL3.5 integrates an InternViT visual encoder with a Qwen3 series language model to support robust image reasoning. These distinct architectures allow us to assess the generalization of our method across different design paradigms.

**Dataset and Benchmarks.** Commonly used benchmarks for visual question answering, such as POPE Li et al. (2023c) and CHAIR Rohrbach et al. (2018), primarily focus on object hallucinations. Since our task aims to alleviate relation hallucinations, we evaluate on two widely used benchmarks specifically designed for this purpose: MMRel Nie et al. (2024) and R-Bench (image-level) Wu et al. (2024). The datasets details are illustrated in Appendix.

**Baselines.** In addition to a standard multimodal large model, we compare our approach with standard Chain-of-Thought prompting Wei et al. (2023) and several training-free methods specifically designed to mitigate relation hallucinations. These include Constraint-Aware Prompting Wu et al.

Table 1: Results on MMRel and R-Bench benchmark. Higher (↑) accuracy, precision, and F1 indicate better performance. The best results are bolded, and the second-best are underlined.

| Model | Method | MMRel | | | R-Bench | | |
|---|---|---|---|---|---|---|---|
| | | Acc | Prec | F1 | Acc | Prec | F1 |
| LLaVA-1.5 | Vanilla | 59.02 | 56.81 | 66.40 | 71.23 | 64.27 | 77.28 |
| | Prompting Wu et al. (2025a) | 62.33 | 60.58 | 68.07 | _75.86_ | 67.28 | _79.60_ |
| | Calibrate Zheng et al. (2024) | _63.50_ | _60.79_ | _70.54_ | 74.28 | _67.86_ | 78.31 |
| | CoT Wei et al. (2023) | 61.88 | 59.14 | 66.25 | 72.91 | 65.88 | 77.87 |
| | Ours | **65.20** | **64.75** | **71.21** | **76.04** | **72.03** | **81.54** |
| InstructBLIP | Vanilla | 57.58 | 55.32 | 66.79 | 69.31 | 62.76 | 76.04 |
| | Prompting Wu et al. (2025a) | _64.52_ | _62.28_ | _71.23_ | _73.65_ | _68.74_ | _80.21_ |
| | Calibrate Zheng et al. (2024) | 62.05 | 61.02 | 69.85 | 72.96 | 66.23 | 78.54 |
| | CoT Wei et al. (2023) | 59.90 | 57.12 | 65.41 | 72.15 | 65.79 | 78.45 |
| | Ours | **65.14** | **64.12** | **74.12** | **75.86** | **70.59** | **81.12** |
| Qwen2.5-VL | Vanilla | 66.10 | 63.07 | 72.14 | 79.85 | 75.19 | 80.88 |
| | Prompting Wu et al. (2025a) | 70.01 | _68.59_ | 75.23 | _81.02_ | _78.38_ | _82.46_ |
| | Calibrate Zheng et al. (2024) | _71.36_ | 67.20 | _76.28_ | 80.98 | 77.05 | 81.52 |
| | CoT Wei et al. (2023) | 67.35 | 65.49 | 72.08 | 80.24 | 76.94 | 81.64 |
| | Ours | **73.52** | **69.65** | **77.45** | **83.92** | **80.23** | **84.63** |
| InternVL3.5 | Vanilla | 71.44 | 67.07 | 75.86 | 82.33 | 78.87 | 83.21 |
| | Prompting Wu et al. (2025a) | 73.01 | 67.65 | 77.12 | _83.97_ | _80.31_ | _84.87_ |
| | Calibrate Zheng et al. (2024) | _74.45_ | _71.46_ | _78.64_ | 82.65 | 79.82 | 83.75 |
| | CoT Wei et al. (2023) | 72.10 | 68.74 | 76.35 | 82.87 | 79.41 | 84.02 |
| | Ours | **75.21** | **72.54** | **78.65** | **85.05** | **82.85** | **85.91** |

(2025a), which relies on text-based prompting, and Detect-then-Calibrate Zheng et al. (2024), which calibrates the final output layer using hidden states from intermediate layers. We chose them because they represent two main ideas in this area: text-prompt-based reasoning and output calibration. Comparing with them allows us to show the benefit of our full text-image reasoning chain in a clear way. The performance of these baseline methods is obtained by re-implementing their experiments on our chosen datasets and experimental setup using the latest public code and their official instructions.

## 4.2 MAIN RESULTS

**Overall Performance.** Table 1 shows ChainMPQ consistently outperforms baselines across both models and benchmarks. On MMRel, ChainMPQ achieves 65.20% accuracy with LLaVA-1.5, improving over the best baseline by 1.7%. Similar improvements are observed with InstructBLIP (65.14% vs 64.52%). On Qwen2.5-VL and InternVL3.5, which already have higher base performance, ChainMPQ also improves accuracy from 71.36% to 73.52% and from 74.45% to 75.21% respectively. The method shows particularly strong precision gains (4.17% higher than the best baseline using LLaVA in R-Bench), indicating reduced false positive relation predictions while simultaneous gains in F1 further confirm that the method improves overall reliability without sacrificing recall.

**Cross-Model Generalization.** ChainMPQ demonstrates consistent improvements across different architectures (LLaVA, InstructBLIP, Qwen-VL and InternVL), suggesting the approach is model-agnostic rather than exploiting specific architectural features.

## 4.3 PERFORMANCE EXPERIMENT

To further improve the efficiency of ChainMPQ, we explored two optimization methods to achieve a trade-off between accuracy and latency.

Table 2: Accuracy and latency comparison of different methods on LLaVA-1.5-7B.

| Benchmark | Method | Acc (%) | Time (s/sample) | $\Delta$Acc / $\Delta$Time |
|---|---|---|---|---|
| MMRel | Vanilla | 59.02 | 0.9 | – |
|  | Full ChainMPQ | 65.20 | 3.3 | 2.58 |
|  | Light1 | 63.78 | 1.5 | **7.93** |
|  | Light2 | 64.25 | 2.1 | 4.36 |
| R-Bench | Vanilla | 71.23 | 0.9 | – |
|  | Full ChainMPQ | 76.04 | 3.3 | 2.00 |
|  | Light1 | 74.50 | 1.5 | **5.45** |
|  | Light2 | 75.08 | 2.1 | 3.21 |

1. **Light1:** We keep only Q1, Q2, and Q5. Since Q1 and Q2 don't need to pass text or visual information to each other, they can run in parallel. When answering Q5, we use the answers of Q1 and Q2 as the text context, and their bias masks M1 and M2, constructed from the attention map, as the visual context.

2. **Light2:** We keep only Q3, Q4, and Q5, and all other settings remain the same. This means the reasoning chain starts from Q3 with no text or visual context, and the remaining steps (from Q3 to Q5) follow the same procedure described in Section 3.5.

The results in Table 2 show that the Light1 has the highest $\Delta$Acc/$\Delta$Time , having a higher accuracy improvement in a shorter time, which means it achieves the best overall trade-off. Therefore, when accuracy is the top priority, we use the full ChainMPQ. When a balance between accuracy and latency is needed, Light1 is the preferred choice.

## 4.4 ABLATION STUDY

We conduct ablation studies to evaluate the impact of key components and hyperparameters in our method, using LLaVA-1.5 on the MMRel benchmark.

### 4.4.1 ABLATION STUDY OF CORE COMPONENTS.

We examine the contributions of ChainMPQ's core components. To assess the role of "Text-guided Attention Enhancement", we remove the enhancement of visual tokens in subject and object regions, such that in the multimodal chain we use $V$ rather than $V'$. To test the importance of the "Construction of Multi-perspective Aware Text Prompts", we replace the five constructed questions with only the relationship question (Q5), meaning that the model only answers Q5 and then the original question. And The model answers the original question using the text context and visual bias produced when answering Q5. To evaluate the contribution of the "Interleaved Text-image Reasoning Chain", we remove the multimodal chain and only use the text context from the previous answer, which means that we keep the entire multi-turn text context, not only the last answer.

As shown in Table 3a, the full ChainMPQ achieves an accuracy of 65.20%. Removing text-guided attention reduces accuracy by 1.14%, suggesting that enhancing subject- and object-related tokens provides limited but measurable benefit in reducing hallucinations. Omitting multi-perspective questions decreases accuracy by 3.68%, confirming their importance in guiding step-by-step inference, which forms the cornerstone of the chain. Eliminating the interleaved chain lowers accuracy by 3.08%, indicating that transferring visual memory plays an equally important role in the reasoning process. Despite these degradations, all three ablated variants still outperform the baseline LLaVA (59.02% accuracy), while the complete ChainMPQ demonstrates the strongest effect.

### 4.4.2 SENSITIVITY ANALYSIS OF HYPERPARAMETERS

We also analyze the sensitivity of $k_{\max}$ , the maximum value for top-K in Eq. 3. We set $k_{\max}$ to 5%, 10%, 30%, and 50% of the total number of patches, corresponding to approximately 10, 20, 70, and 120 tokens. We also vary the maximum bias weight parameter $\lambda$, which is the coefficient of $\alpha$ in Eq. 5, and test values of 3, 5, and 7. As shown in Figure 3b, performance peaks at $\lambda = 5$

| Model Variants | Acc | Prec | F1 |
|---|---|---|---|
| LLaVA-1.5 | 59.02 | 56.81 | 66.40 |
| **ChainMPQ(Full)** | **65.20** | **64.75** | **71.21** |
| w/o Enhancement | 64.06 | 63.25 | 69.42 |
| w/o Multi-perspective | 61.52 | 60.84 | 67.53 |
| w/o Interleaved | 62.12 | 61.47 | 68.01 |

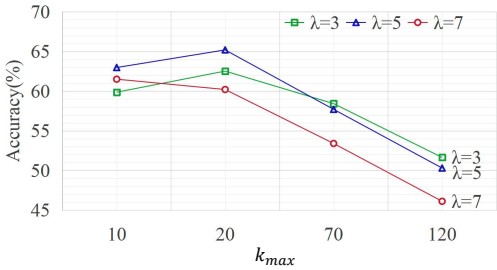

(a) Ablation study with different model variants.

(b) Sensitivity of Accuracy to $k_{\max}$ and $\lambda$.

Figure 3: Ablation Results on MMRel using LLaVA-1.5: (a) Ablation study; (b) Sensitivity analysis.

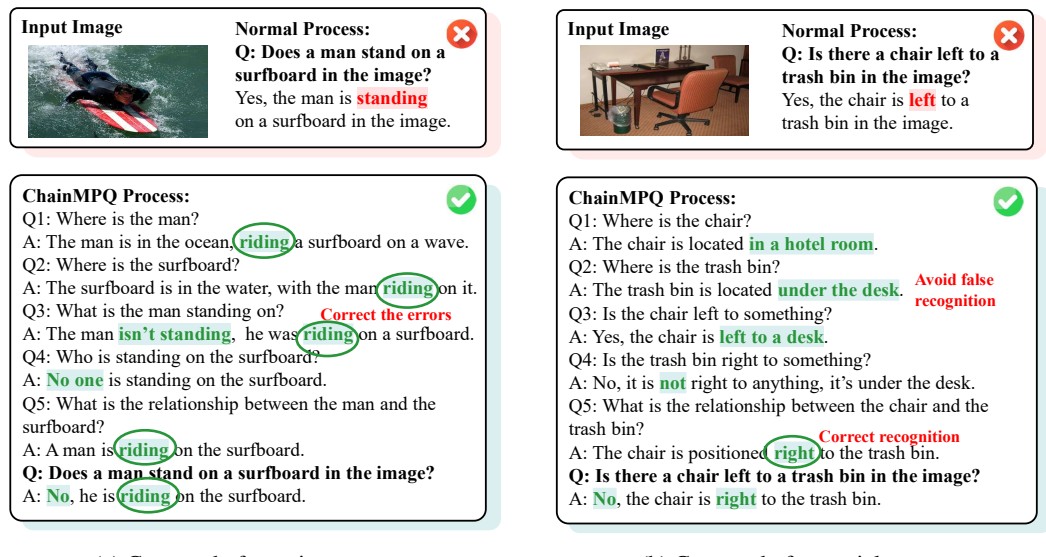

(a) Case study for action category

(b) Case study for spatial category

Figure 4: The full ChainMPQ answering process vs. directly output

and $k_{\max} = 20$, where the accuracy reaches 65.20%. When $k_{\max}$ is too large, the model uses almost all image tokens, which weakens its focus on important regions and breaks the normal attention process. In contrast, when $k_{\max}$ is too small, the model may miss key information, especially for questions that rely on scattered visual features. Similarly, if $\lambda$ is too large, the model becomes overly dependent on past memory, which disturbs attention propagation and reduces accuracy. If $\lambda$ is too small, the bias is weak and the performance is close to the original LLaVA baseline.

## 4.5 CASE STUDY

To intuitively illustrate the ChainMPQ process, we present two real examples from MMRel, namely an action case and a spatial case, which together account for over 90% of the MMRel dataset Nie et al. (2024). Due to page limits, another case study of competitive type is provided in Appendix A.6.2.

Figure 4 illustrates the complete processing pipeline of ChainMPQ. In the action case "Does a man stand on a surfboard in the image?", the baseline answers "yes", confusing the true relation "riding" with "standing". ChainMPQ first localizes the subject and the object with two guiding questions, identifies the action "riding," and propagates this cue through textual context and attention transfer to produce the correct answer "no". In the spatial case "Is there a chair left to a trash bin in the image?", the baseline is again incorrect. ChainMPQ retrieves the locations of the chair and the bin under the desk, constrains subsequent reasoning to the correct regions, and then verifies the directional relation to conclude that the chair is to the "right" of the bin. Across both cases, decomposition helps the

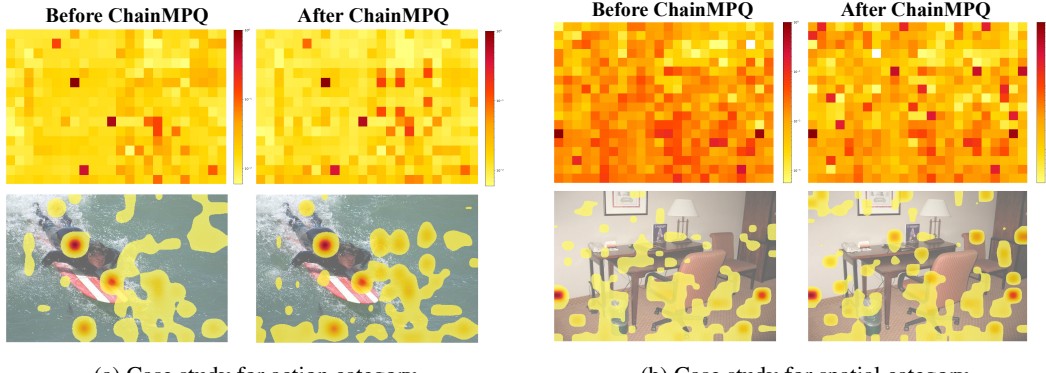

(a) Case study for action category          (b) Case study for spatial category

Figure 5: Comparison of attention maps between directly answering the original question and answering it using ChainMPQ.

model locate the correct regions, reduce false language priors, and follow a clear path to verify relations, leading to answers that aligned with the image.

Figure 5 presents that across both action and spatial cases, ChainMPQ produces attention maps that are more sharply concentrated on task-relevant regions, especially the subject and the object. This pattern indicates that the focused attention triggered by earlier related subquestions is propagated through the chain via textual context and attention transfer, so that when answering the original question the model attends more precisely to the related areas. In both examples, ChainMPQ can suppress attention to irrelevant background areas and focus attention on the interaction between the subject and the target object, aligning the attention trajectory with the queried relation. The resulting focus correlates with corrected predictions and fewer contradictions between the image and the answer, which suggests improved grounding and more faithful relational inference.

## 5  CONCLUSION

We propose **ChainMPQ** (**M**ulti-**P**erspective **Q**uestions guided Interleaved Text-image Reasoning **Chain**), a training-free framework that enhances the ability of LVLMs to recognize relationships by utilizing accumulated textual and visual memories. ChainMPQ first strengthens the relevant visual tokens using keywords extracted from the question. It then constructs a set of complementary questions from different perspectives, each closely centered on the subject, the object, and their relationship. These questions are sequentially fed into the model, with both textual and visual memories from earlier rounds propagated to subsequent ones, thereby forming an interleaved chain of image and text. Experimental results demonstrate that ChainMPQ effectively reduces relation hallucinations and improves factuality across multiple LVLMs and benchmarks, providing a simple yet robust framework for step-by-step relational inference.

## 6  FUTURE WORK

Although ChainMPQ consistently improves the mitigation of relation hallucinations, it still relies on attention distributions as a proxy for visual evidence, which may not fully capture the underlying reasoning process. Future work will incorporate a causality-based attribution mechanism. By detecting and correcting outputs that contradict the image, it prevents errors from cascading.

It is worth mentioning that a major challenge in relation hallucination lies in the spatial granularity of visual tokens, which often misalign with real-world object boundaries. This misalignment, observed in lower performance for the spatial category in MMRel in our experiments, can hinder accurate relational reasoning. Future research may alleviate this issue by exploring multi-scale visual representations or by integrating explicit scene graph representations to improve robustness in fine-grained relational understanding.

ACKNOWLEDGMENTS

The work is partially supported by the U.S. National Science Foundation (NSF) Grant CRII 2451683, an NVIDIA Academic Grants Program, a U.S. Bank Academic Research Award, the University of California, Merced, and a UC Merced Faculty Research Award. The views and conclusions are those of the authors and do not necessarily reflect the official policy or position of the U.S. Government.

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

# A APPENDIX

## A.1 DETAILED ALGORITHM

Algorithm 2 describes the complete execution process of ChainMPQ in detail.

---

**Algorithm 2:** ChainMPQ (training-free, interleaved text–image reasoning)

---

**Input:** Image $I$, relational question $Q$
**Output:** Final answer $A$
**Retained Intermediates:** $(\text{Attn}_i, M_i)$
**Symbols:** Visual tokens $V$, enhanced visual tokens $V'$, complementary questions $Q_{1:5}$, textual memory
$\quad\quad\mathcal{T}$ (accumulates $(Q_i, A_i)$), visual memory $\mathcal{V}$ (accumulates $M_i$).
**Hyperparams:** $\lambda$, $k_{\max}$

**STEP I: Text-guided Attention Enhancement**
1. Extract subject/object keywords $K$ from $Q$.
2. $V \leftarrow \text{VisualEncoder}(I)$, $\quad X \leftarrow \text{TextEncoder}(K)$.
3. Use cross attention on $X$ and $V$ to enhance visual tokens .

**STEP II: Multi-Perspective Prompt Construction**
4. Extract subject $[S]$, object $[O]$, relation $[R]$ from $Q$.
5. Build five complementary questions $Q_{1:5}$ with mask strategy:
$\quad Q_1$: Where is [S]?
$\quad Q_2$: Where is [O]?
$\quad Q_3$: What is [S] [R]? (mask object)
$\quad Q_4$: What is [R] [O]? (mask subject)
$\quad Q_5$: What is the relationship between [S] and [O]? (mask relation)
6. Initialize $\mathcal{T} \leftarrow \varnothing, \mathcal{V} \leftarrow \varnothing$.;

**STEP III: Interleaved Text–Image Reasoning**
7. Direct answers for $Q_1, Q_2$ (no context, no bias):
$\quad$ For $i \in \{1, 2\}$: $\quad A_i \leftarrow Q_i$ using $V'$; $\mathcal{T} \leftarrow \mathcal{T} \cup \{(Q_i, A_i)\}$
8. Context- and mask-aware answers for $Q_3$–$Q_5$:
$\quad$ For $i = 3..5$:
$\quad\quad$ (a) Compute aggregated attention $\text{Attn}_i(\lambda)$.
$\quad\quad$ (b) Form top-$k$ set and build $M_i(k_{\max})$:
$\quad\quad$ (c) Decode with textual & visual memory:
$\quad\quad\quad A_i \leftarrow Q_i$ using $V', \mathcal{T}; M_i, \mathcal{T} \leftarrow \mathcal{T} \cup \{(Q_i, A_i)\}; \mathcal{V} \leftarrow \mathcal{V} \cup \{M_i\}$.
9. Final answer: $A \leftarrow Q, \mathcal{T}, \mathcal{V}$.

---

## A.2 DATASET DETAILS

**MMRel:** A vision-language benchmark specifically designed to assess and improve large multi-modal models' understanding of inter-object relationships. approximately 10.14K Yes/No question–answer pairs and covers three relation types: spatial, action, and comparative. MMRel incorporates both real-world dataset(Visual Genome) and synthetic images (via SDXL and DALL·E), allowing for diverse relational reasoning scenarios, including adversarial and counter-intuitive cases. The dataset is constructed through a combination of GPT-4V–assisted annotations and human validation to ensure high-quality labels. Beyond evaluation, MMRel serves as a valuable resource for instruction tuning and fine-tuning, showing clear improvements in relational grounding when used for training. It reveals that many current vision-language models rely heavily on linguistic priors, often hallucinating relations not supported by visual evidence.

**R-Bench:** A diagnostic benchmark focused on detecting relationship hallucinations in large vision-language models (LVLMs). It comprises image-level and instance-level yes/no questions that test a model's ability to identify whether specific relations between objects truly exist. The benchmark categorizes hallucinations into three co-occurrence types: relation–relation, subject–relation, and relation–object to isolate sources of error. R-Bench samples are sourced primarily from NoCaps, ensuring minimal training overlap. Experiments show that LVLMs frequently over-rely on commonsense priors, misjudging relations in visually ambiguous or adversarial contexts. As a result, R-Bench is especially effective for fine-grained analysis of relational reasoning errors, offering insights into model biases and areas for improvement. We clarify that our experiments use the image-

level setup of R-Bench. The image-level and instance-level setups share almost the same images and questions, but the instance-level setup also provides bounding box coordinates and focuses more on the relationship between specific object instances. Since most datasets do not provide bounding boxes, and locating the objects involved in a relation is itself an essential ability of LVLMs, we choose the image-level setup for our main experiments to give a fair and clear evaluation of how ChainMPQ improves relational understanding.

## A.3 IMPLEMENTATION DETAILS OF MULTI-PERSPECTIVE QUESTION GENERATION

First, we use an NLP tool to extract the subject $S$, object $O$, and relation $R$ from the original question. For example, the question "Does the dog chase a disc in the image?" can be decomposed into the subject "dog", the object "disc", and the relation "chase". We then build two questions to locate the subject and the object. In this example, they are "Q1: Where is a dog in the image?" and "Q2: Where is a disc in the image?"

Next, we apply a masking strategy. In each step, one of the three elements is masked, and the other two are used to form a new question. This helps the model focus on the remaining elements and check the original relation from different views. For this example, after masking and reconstruction, we obtain three new questions: "Q3: What is the dog chasing?", "Q4: What is the disc being chased by?", and "Q5: What is the relationship between the dog and the disc?" We use GPT-3.5 Turbo to refine each question so that the wording is clear and grammatically correct.

## A.4 EXPERIMENT IMPLEMENTATION DETAILS

In the **Construction of Multi-Perspective Aware Text Prompts** step, we employ the spaCy NLP toolkit to extract salient keywords from the input questions. These keywords are then used to refine the generated prompts, ensuring that the decomposed questions remain semantically coherent and closely aligned with the original query.

In the **Construction of the Interleaved Chain of Image and Text** step, we set $n$ as 3 to capture the last 3 layers' attention in decoder. We set $k_{max}$ to 10% of the total number of visual patches (i.e., $k_{max} = 20$). This proportion provides a balance between capturing sufficient visual context and avoiding noise from irrelevant regions. The maximum bias weight parameter $\lambda$ (the coefficient of $\alpha$) is fixed at 5, which we found to be a stable value across different datasets and models.

Regarding the comparative experiments with other methods, we reproduced both baseline methods using the latest public code and their official instructions. We only replaced the base models with LLaVA-1.5-7B and InstructBLIP-7B, and we changed the datasets to MMRel and R-Bench to match our setting. All comparisons are made under a fully fair setup. For Constraint-Aware Prompting Wu et al. (2025a), we used the combined-prompt, which is Prompt 10, as it gives the best accuracy. For Detect-then-Calibrate Zheng et al. (2024), we ran their released scripts without any change to the parameters.

## A.5 EXPERIMENTS ON GENERAL REASONING BENCHMARKS

To explore whether improved relation reasoning also lead to better general reasoning or broader application outcomes. We also evaluated ChainMPQ on two general multimodal benchmarks, MM-Bench Liu et al. (2024c) and MME Fu et al. (2025). And the results show that ChainMPQ brings consistent gains not only in relation reasoning but also in broader multimodal abilities.

These tasks cover a broad range of multimodal abilities, such as visual logic reasoning, scene understanding, OCR, and object grounding, and are not limited to relation reasoning. Specifically, MMBench defines three capability levels and twenty fine-grained ability dimensions for evaluating different reasoning and perception skills of MLLMs. MME includes tests for both perception and cognition. The perception part covers coarse- and fine-grained skills such as object existence, counting, location, color recognition, poster recognition, celebrity recognition, scene and landmark identification, and artwork recognition. The cognition part includes commonsense reasoning, numerical reasoning, translation, and code reasoning. It contains fourteen sub-tasks in total.

The MMBench results are shown in Table 3. Here Overall is the weighted average over all sub-tasks and is the most important indicator of general ability. RR (Relation Reasoning) refers to relation

Table 3: **MMBench** results (Overall & Relation Reasoning). Higher is better.

| Model | Overall ↑ | ΔOverall | RR ↑ | ΔRR |
|---|---|---|---|---|
| LLaVA-v1.5-7B | 66.5 | – | 58.8 | – |
| + ChainMPQ | 67.8 | +1.3 | 61.3 | +2.5 |
| InstructBLIP-7B | 63.9 | – | 52.5 | – |
| + ChainMPQ | 65.5 | +1.6 | 55.2 | +2.7 |
| Qwen2.5-VL-7B | 83.2 | – | 78.2 | – |
| + ChainMPQ | 84.7 | +1.5 | 81.8 | +3.6 |
| InternVL3-8B | 83.6 | – | 82.5 | – |
| + ChainMPQ | 84.2 | +0.6 | 83.9 | +1.4 |

Table 4: **MME** results (Overall, Perception, Cognition). Higher is better.

| Model | Overall ↑ | ΔOverall | Perception ↑ | ΔPerc | Cognition ↑ | ΔCog |
|---|---|---|---|---|---|---|
| LLaVA-v1.5-7B | 1808.4 | – | 1506.2 | – | 302.1 | – |
| + ChainMPQ | 1898.3 | +89.9 | 1545.6 | +39.4 | 352.7 | +50.6 |
| InstructBLIP-7B | 1391.4 | – | 1137.1 | – | 254.3 | – |
| + ChainMPQ | 1484.6 | +87.2 | 1176.0 | +38.9 | 308.6 | +54.3 |
| Qwen2.5-VL-7B | 2312.1 | – | 1698.1 | – | 613.9 | – |
| + ChainMPQ | 2350.8 | +38.7 | 1719.2 | +21.1 | 631.5 | +17.6 |
| InternVL3-8B | 2422.0 | – | 1748.4 | – | 673.6 | – |
| + ChainMPQ | 2457.1 | +35.1 | 1771.4 | +23.0 | 685.7 | +12.1 |

reasoning. The MME results are shown in Table 4. Here Overall is the average score across all fourteen tasks. Perception measures detection, counting, color, position, and OCR skills, where we do not expect large gains.Cognition measures high-level semantic understanding such as reasoning, description, and complex question answering.

In sum, these results show that improving relation reasoning does lead to better general reasoning. ChainMPQ brings consistent gains on the Overall scores of both MMBench and MME, especially in the Relation Reasoning and Cognition part, confirming that stronger relation understanding supports broader multimodal reasoning abilities.

## A.6  DETAILED CASE STUDY

### A.6.1  ATTENTION ANALYSIS IN CASE STUDY

As shown in Figure 6, ChainMPQ progressively guides the model's reasoning from object localization to relational understanding. The first two sub-questions help the model correctly identify the man and the surfboard, and the attention maps show a clear shift toward the regions that match each answer. The following three sub-questions gradually move the attention toward the interaction between these objects, which helps the model understand the relationship more clearly.

Compared with the direct-answer setting, where the attention maps are scattered and often point to irrelevant areas, ChainMPQ produces more focused and meaningful patterns. In particular, for the question "What is the man standing on", the attention highlights the surfboard instead of the water. This prevents the model from being confused by the wording of the original question.

This case shows that ChainMPQ not only breaks down a complex question into simple and clear steps but also adjusts the attention process in a helpful way. By guiding the model step by step, ChainMPQ improves both the reasoning path and the final answer, which leads to more stable and accurate relational predictions.

### A.6.2  CASE STUDY FOR COMPETITIVE CATEGORY IN MMREL

We only showed two case studies in the main text because of space limits. This does not mean that our method works better on action or spatial questions than on other types. ChainMPQ is also

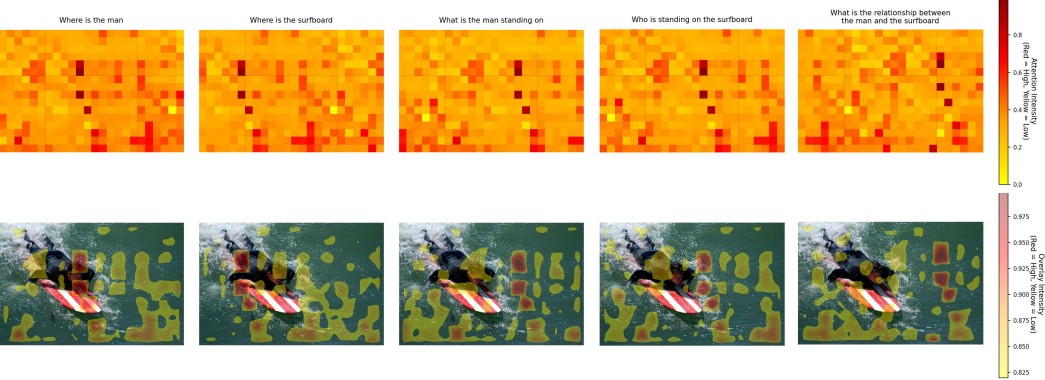

Figure 6: For each generated question, the attention map during the model's answer generation process using ChainMPQ is shown.

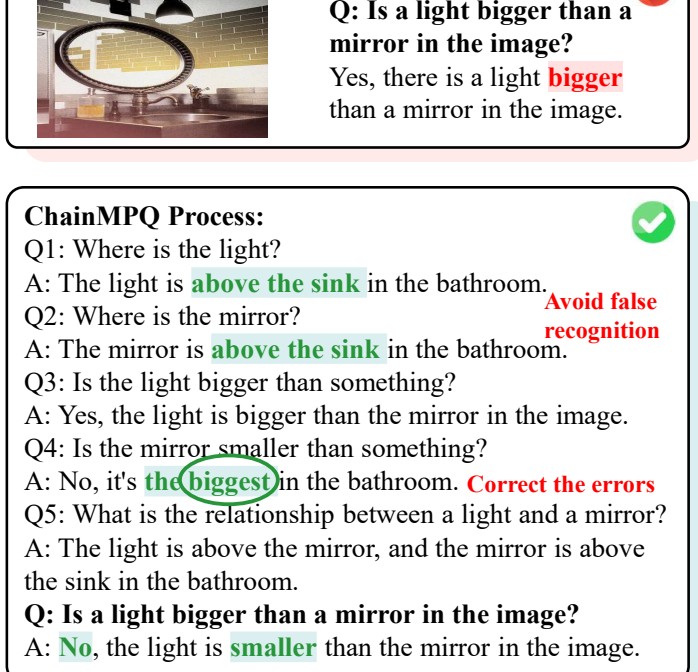

Figure 7: Case study for competitive category

effective on competitive cases. In Figure 7, we present an example from the competitive domain. This case shows that ChainMPQ can handle competitive questions with the same level of stability as action and spatial questions. It also suggests that our method is not restricted to one type of relation and can work well across different scenarios.

