# OpenReview forum: "ChainMPQ: Interleaved Text-Image Reasoning Chains for Mitigating Relation Hallucinations"
_ICLR.cc/2026/Conference — ICLR 2026 Poster_

### Official Review · Reviewer_HnMX · 2025-10-15

**Soundness:** 4
**Presentation:** 4
**Contribution:** 4
**Rating:** 8
**Confidence:** 5

**Summary:**

The paper introduces ChainMPQ, a novel method designed to mitigate relation hallucinations in large vision-language models (LVLMs). Relation hallucinations, a type of error in multimodal reasoning, occur when the model fails to accurately infer relationships between recognized objects. The proposed method works by first extracting subject and object keywords, enhancing the visual tokens associated with these entities, and then constructing multiple interleaved questions focusing on different components of a relationship. These questions guide the model step-by-step, leveraging both textual and visual memory to progressively refine relational inference. The paper demonstrates the effectiveness of ChainMPQ through experiments on multiple LVLMs and benchmarks, showing significant improvements in reducing relation hallucinations.

**Strengths:**

1. The decomposition of relational reasoning into multiple, focused questions is a unique approach. By systematically addressing each component of a relationship (subject, object, and relation), ChainMPQ offers a more structured and interpretable reasoning process compared to previous methods.

2. The method does not require additional training or fine-tuning, making it a practical solution for mitigating hallucinations in pre-trained models. This is a key advantage in real-world applications where retraining large models may not be feasible.

3. The experiments show consistent improvements across different LVLMs (LLaVA-1.5 and InstructBLIP) and benchmarks (MMRel and R-Bench), demonstrating that ChainMPQ is effective in various contexts.

4. The method’s success across different architectures further validates its generalizability and robustness.

**Weaknesses:**

1. There are other training-free methods[1,2] that adjust attention maps to enhance MLLM performance. These works should be cited and discussed in comparison to ChainMPQ to provide a broader context for the proposed method.

2. R-Bench contains both image-level and instance-level setups. Which setup is used for the experiment in Table 1? If the results in Table 1 are based on the image-level setup, can this method also perform well in the instance-level setup?

3. The current baselines only include LLaVA-1.5 and InstructBLIP. Given the rapid advancements in LVLMs, it would be beneficial to include newer models, such as Qwen2.5-VL[3] and InternVL3.5[4], to demonstrate the method’s effectiveness on state-of-the-art models.

[1] Controlmllm: Training-free visual prompt learning for multimodal large language models

[2] MLLMs Know Where to Look: Training-free Perception of Small Visual Details with Multimodal LLMs

[3] Qwen2.5-VL Technical Report

[4] Internvl3. 5: Advancing open-source multimodal models in versatility, reasoning, and efficiency

**Questions:**

See weakness.

---

> ### Author Response · Authors · 2025-11-25
> **Reply to weaknesses (1/2)**
>
> Thank you very much for your positive comments on our multi-perspective relational reasoning framework, and for recognizing the practicality and stable performance of our method across different models. Your feedback is very encouraging. We have added more experiments based on your suggestions, which further demonstrate the strong effectiveness of ChainMPQ.
>
> **Q1:** There are two other training-free methods that adjust attention maps to enhance MLLM performance. Can the authors compare ChainMPQ with these methods?
>
> [1] Controlmllm: Training-free visual prompt learning for multimodal large language models
>
> [2] MLLMs Know Where to Look: Training-free Perception of Small Visual Details with Multimodal LLMs
>
> **A1:** Although these two methods also modify attention maps and improve MLLM performance, they are not designed for relation hallucination. This is why we did not include them at first. But we have now added experiments comparing our method with these two approaches, and we also include the newer backbone models Qwen2.5-VL and InternVL3.5 as you suggested. The results show that ChainMPQ still achieves the strongest performance across all models.
>
> Table 1. Comparison with Attention-Based Training-Free Methods on **MMRel**.
>
> | Model               | Vanilla | Control-MLLM | Where2Look | **ChainMPQ (Ours)** |
> | ------------------- | ------- | ------------ | ---------- | ------------------- |
> | **LLaVA-1.5-7B**    | 59.02   | 61.36        | 61.01      | 65.20               |
> | **InstructBLIP-7B** | 57.58   | 60.19        | 63.42      | 65.14               |
> | **Qwen2.5-VL-7B**   | 66.10   | 69.70        | 68.26      | 73.52               |
> | **InternVL3.5-8B**  | 71.44   | 73.45        | 74.14      | 75.21               |
>
> Table 2. Comparison with Attention-Based Training-Free Methods on **R-Bench**
>
> | Model               | Vanilla | Control-MLLM | Where2Look | **ChainMPQ (Ours)** |
> | ------------------- | ------- | ------------ | ---------- | ------------------- |
> | **LLaVA-1.5-7B**    | 71.23   | 74.04        | 74.52      | 76.04               |
> | **InstructBLIP-7B** | 69.31   | 70.65        | 70.88      | 75.86               |
> | **Qwen2.5-VL-7B**   | 79.85   | 81.77        | 82.55      | 83.92               |
> | **InternVL3.5-8B**  | 82.33   | 83.21        | 83.45      | 85.05               |
>
>
>
> **Q2:** R-Bench contains both image-level and instance-level setups. Which setup is used for the experiment in Table 1?
>
> **A2:** Thank you for the question. The results in Table 1 are based on the image-level setup of R-Bench, which we will clarify in the revision.
>
> We chose the image-level setup for the experiments based on the following considerations:
> The images and questions in the image-level and instance-level setups are almost the same, but the instance-level setup also provides bounding box coordinates and focuses more on the relationship between specific objects. However, most datasets do not offer bounding box annotations, and being able to locate the objects involved in a relation is exactly the ability an LVLM needs when understanding relations. To keep the evaluation fair and to clearly show how ChainMPQ improves the model’s relation understanding ability, we used the image-level setup in the main experiments.
>
> And we have clarified this point in the revised version of the paper.
>
> We have also added experiments under the instance-level setup. The results show that ChainMPQ performs well in this setup as well, which further supports the robustness of our method.
>
> Table 3. Performance on R-Bench Instance-level (Box)
>
> | **Model**           | **Method**               | **R-Bench (Instance-level)** |           |         |
> | ------------------- | ------------------------ | ---------------------------- | --------- | ------- |
> |                     |                          | **Acc↑**                     | **Prec↑** | **F1↑** |
> | **LLaVA-1.5-7B**    | Vanilla                  | 53.15                        | 51.71     | 67.10   |
> |                     | Prompting (Wu et al.)    | 57.41                        | 55.24     | 68.04   |
> |                     | Calibrate (Zheng et al.) | 55.15                        | 52.05     | 65.47   |
> |                     | **ChainMPQ (Ours)**      | 59.03                        | 57.41     | 69.47   |
> | **InstructBLIP-7B** | Vanilla                  | 51.95                        | 51.14     | 64.52   |
> |                     | Prompting (Wu et al.)    | 55.21                        | 53.74     | 66.48   |
> |                     | Calibrate (Zheng et al.) | 54.86                        | 52.41     | 67.41   |
> |                     | **ChainMPQ (Ours)**      | 57.04                        | 54.79     | 69.73   |

---

> ### Author Response · Authors · 2025-11-25
> **Reply to weaknesses (2/2)**
>
> **Q3:** The paper shows good results on LLaVA-1.5 and InstructBLIP. Can you also add experiments on state-of-the-art models such as Qwen2.5-VL and InternVL3.5?
>
> **A3:** Thank you for the suggestion. We have added experiments on Qwen2.5-VL-7B and InternVL3.5-8B. The results show that ChainMPQ still brings clear gains on these stronger backbone models, which confirms that our method remains effective even when the base model already has advanced reasoning ability.
>
> Table 4. Comparison on MMRel and R-Bench (Acc / Prec / F1).
>
> | **Model**       | **Method**                    | **MMRel** Acc↑ | Prec↑ | F1↑   | **R-Bench** Acc↑ | Prec↑ | F1↑   |
> | --------------- | ----------------------------- | -------------- | ----- | ----- | ---------------- | ----- | ----- |
> | **Qwen2.5-VL**  | Vanilla                       | 66.10          | 63.07 | 72.14 | 79.85            | 75.19 | 80.88 |
> |                 | Prompting (Wu et al. 2025a)   | 70.01          | 68.59 | 75.23 | 81.02            | 78.38 | 82.46 |
> |                 | Calibrate (Zheng et al. 2024) | 71.36          | 67.20 | 76.28 | 80.98            | 77.05 | 81.52 |
> |                 | **ChainMPQ (Ours)**           | 73.52          | 69.65 | 77.45 | 83.92            | 80.23 | 84.63 |
> | **InternVL3.5** | Vanilla                       | 71.44          | 67.07 | 75.86 | 82.33            | 78.87 | 83.21 |
> |                 | Prompting (Wu et al. 2025a)   | 73.01          | 67.65 | 77.12 | 83.97            | 80.31 | 84.87 |
> |                 | Calibrate (Zheng et al. 2024) | 74.45          | 71.46 | 78.64 | 82.65            | 79.82 | 83.75 |
> |                 | **ChainMPQ (Ours)**           | 75.21          | 72.54 | 78.65 | 85.05            | 82.85 | 85.91 |

---

### Official Review · Reviewer_MyiZ · 2025-10-25

**Soundness:** 2
**Presentation:** 2
**Contribution:** 2
**Rating:** 4
**Confidence:** 3

**Summary:**

The goal of this work is to tackle relational hallucination in LVLMs which is under explored in previous literature in comparison to object and attribute based hallucinations. The proposed method is called ChainMPQ, which is essentially a training free pipeline that converts subject and object keywords from the prompt into a series of questions about the subject, object, and their relation, and uses these to guide cross-attention via an interleaved chain-of-thought mechanism. The goal is to sharpen attention on relation-relevant regions and reduce errors without additional training.

The main findings of the paper show that across two different model families, ChainMPQ is able to reduce relational hallucination on two benchmarks that measure relation hallucinations. More ablation studies are provided to show which types of the questions contribute to the overall improvement of the final results. Their main finding here states that multi-perspective questions and the interleaved chain are the biggest contributors to performance gain.

**Strengths:**

- Interesting application of interleaved chain-of-thought prompting to reduce relational hallucinations. The staged prompts around subject, object, and relation give the model clear subgoals and make the intervention easy to understand.
- Results clearly demonstrate the benefit of this approach on relational hallucination benchmarks. The ablations help isolate where gains come from, e.g. multi-stage enhancement of visual tokens seems to help the most.
- Training free and low cost to deploy, which makes the method practical for settings where fine-tuning is not feasible.

**Weaknesses:**

- Organization and detailing of content can be improved a lot. For instance, implementation details of question generation from the text prompt are not included in the main text. Similarly, the baselines are not described in detail (trained/training-free, how does their approach differ from ChainMPQ, etc). In the ablations section, the description of ablations was not very clear to me (e.g. "five constructed questions with a single relationship question" -- how is this constructed?, "we remove the multimodal chain and only use the text context from the previous answer." -- is this the entire multi-turn text context or just the last turn answer?). The paper would benefit by including these details/examples in the main/appendix sections.
- I think the multi perspective question generation aspect of this work is interesting, but currently lacks a general framework for the benefit of future practitioners who might want to extend this work to new benchmarks or new domains. Some discussion or a framework around that can help enhance the generality of this work.
- Contributions seem siloed around relational hallucinations so the novelty seems limited. The work can be improved by showing how this module complements more general hallucination mitigation strategies.

**Questions:**

- Can the authors elaborate on the questions raised above on distinguishing details about the ablations: (i) replaces five constructed question with a single question, and (ii) removal of multimodal chains that was raised in the Weaknesses section?
- The case study section says "we present two real examples from MMRel, namely an action case and a spatial case, which together account for over 90% of the MMRel dataset" -- are there any experiments to check the diversity of cases in the two datasets beyond action and spatial? My worry is that this question construction approach is more geared towards the templates most suitable for action spatial domains.
- Can the authors highlight how the exact differences between ChainMPQ and the baselines compared (Constraint-Aware Prompting and Detect-then-Calibrate) and the rationale behind choosing these over other related works?

---

> ### Author Response · Authors · 2025-11-25
> **Reply to weaknesses (1/2)**
>
> Thank you very much for your positive comments on our interleaved vision–language reasoning chain and the strong performance of our method. We have made the following additions and improvements based on your suggestions.
>
> **W1.1：** Give the execution details of generating multi-perspective questions
>
> **R1.1：** Thank you for the suggestion. We have added the full execution details of generating multi-perspective questions in the paper:
>
> First, we use an NLP tool to extract the subject *S*, object *O*, and relation *R* from the original question. For example, the question “Does the dog chase a disc in the image?” can be decomposed into the subject “dog”, the object “disc”, and the relation “chase”. We then build two questions to locate the subject and the object. In this example, they are “Q1: Where is a dog in the image?” and “Q2: Where is a disc in the image?”.
>
> Next, we apply a masking strategy. In each step, one of the three elements is masked, and the other two are used to form a new question. This helps the model focus on the remaining elements and check the original relation from different views. For this example, after masking and reconstruction, we obtain three new questions: “Q3: What is the dog chasing?”, “Q4: What is the disc being chased by?”, and “Q5: What is the relationship between the dog and the disc?”. We use GPT-3.5 Turbo to refine each question so that the wording is clear and grammatically correct.
>
> **W1.2：** Are the baselines trained or training-free, and how are they different from ChainMPQ?
>
> **R1.2：** As stated in the second paragraph of the Introduction, both Prompting (Wu et al.) and Calibrate (Zheng et al.) are **training-free**. They treat relation reasoning as a single-step process and expect the model to identify entities and predict the relation at the same time. ChainMPQ is different because it introduces a step-by-step reasoning process and makes full use of both text and visual information during reasoning. More specifically, Prompting (Wu et al.) only uses text prompt templates and does not use visual features. Calibrate (Zheng et al.) adjusts the final logits with intermediate features but does not guide the model to reason step by step.
>
> **W1.3:** In the ablation experiment, does “only use the text context from the previous answer” refer to the entire multi-turn text or only the last answer?
>
> **R1.3:** "only use the text context from the previous answer” refers to the **entire multi-turn text context**, not only the last answer.
>
> According to your suggestion, we have added basic descriptions of the baselines in the experimental setup and clarified the operations of each ablation in the ablation section.
>
>
>
> **W2：** I find the multi-perspective question generation part very interesting. It would be even better if this work could be extended to new benchmarks or new domains through a more general framework.
>
> **R2：** Thank you for the suggestion. The multi-perspective question generation module is not limited to relation questions. The key idea is to break a complex vision-language problem into several simple and complementary sub-questions. In relation reasoning, we decompose the question into the subject [S], object [O], and relation [R]. For other tasks, we can follow the same idea by identifying the core elements of the task and building sub-questions based on them.
>
> We summarize a general framework for extending this module to other tasks:
>
> **1. Identify the core components of the task.**
>  For example:
>
> - Attribute reasoning → (object, attribute type, attribute value)
> - Action recognition → (agent, action, scene)
>
> **2. Generate complementary questions.**
> Based on the core components, we can create localization questions and **missing-element masking questions**. Each core element can be checked through one focused sub-question. This way, tasks such as attribute reasoning, action recognition, spatial reasoning, counting, and even medical VQA can be handled by forming different combinations of these sub-questions.
>
> This shows that the idea of multi-perspective question generation can be applied to many new benchmarks and domains through a simple and general extension strategy.

---

> ### Author Response · Authors · 2025-11-25
> **Reply to weaknesses (2/2)**
>
> **W3:** ChainMPQ works well for relation hallucination. Can it also be connected to more general hallucination mitigation strategies?
>
> **R3:** Thank you for the question. Although ChainMPQ is designed for relation hallucination, its core idea is a general form of multi-step grounding.  ChainMPQ is a training-free and plug-and-play framework, so it can be combined with many general hallucination mitigation methods. For example:
>
> **• Combination with Contrastive Decoding.**
>  ChainMPQ produces a multi-step reasoning chain and gives a visual-enhanced logit path. Contrastive Decoding can then compare this path with the vanilla path to enforce visual consistency.
>
> **• Combination with Generate-then-Verify methods.**
> These methods first generate an answer, then use an extra module to check visual consistency. If the answer is not consistent with the image, they resample or adjust the logits. ChainMPQ can be used before the verification stage to give a more reliable relation understanding, which makes the later verification more stable.
>
> **• Combination with DPO, GRPO, or other RL-based correction.**
> These methods reduce hallucination during the fine-tuning stage. ChainMPQ can be applied after fine-tuning, since it does not change model weights. ChainMPQ can also serve as a preference data generator. The corrected answers from ChainMPQ can be used to build high-quality (preferred, dispreferred) pairs to improve RL training.
>
> These examples show that ChainMPQ is compatible with general hallucination mitigation strategies.

---

> ### Author Response · Authors · 2025-11-25
> **Reply to questiones**
>
> **Q1:** Distinguish details about the ablations:
>
> (i) replacing five constructed questions with a single question, and
> (ii) removal of multimodal chains
>
> **A1:**
>
> (i) Replacing five constructed questions with a single question.
> As described in Section 4.3.1 “Ablation Study of Core Components”, this ablation tests the importance of constructing multi-perspective questions. We replace all five constructed questions with only one question about the relation (the fifth constructed question).
>
> In this setting, the model no longer follows a multi-step question chain. It only answers two questions: Q5 and the original question. The answer to Q5 is used as the text context, and we build the visual bias. The model then answers the original question based on both the text and visual information from Q5.
>
> (ii) Removal of multimodal chains.
> As stated in A1.3, this means that we keep the entire multi-turn text context, not only the last answer.
>
>
>
> **Q2：** In the case study, the examples are from the action and spatial domains. Does this mean the method works better on action or spatial cases?
>
> **A2：** Thank you for the question and we understand your concern. We only showed two examples due to space limits, but this does not mean that our method is more suitable for action or spatial questions. In fact, ChainMPQ also performs well on competitive cases. Below we provide an example from the competitive domain.
>
> Image: https://cs.stanford.edu/people/rak248/VG_100K_2/2400006.jpg
>
> Original question: Is a light bigger than a mirror in the image?
>
> Ground truth: No.
>
> Direct answer: Yes, there is a light bigger than a mirror in the image.
>
> **ChainMPQ reasoning:**
>
> Q1: Where is the light?
>
> A1: The light is above the sink in the bathroom.
>
> Q2: Where is the mirror?
>
> A2: The mirror is above the sink in the bathroom.
>
> Q3: Is the light bigger than something?
>
> A3: Yes, the light is bigger than the mirror in the image.
>
> Q4: Is the mirror smaller than something?
>
> A4: No, it's the biggest in the bathroom. (**correct the previous error**)
>
> Q5: What is the relationship between a light and a mirror?
>
> A5: The light is above the mirror, and the mirror is above the sink in the bathroom.
>
> Original question: Is a light bigger than a mirror in the image?
>
> Final answer: No, the light is smaller than the mirror in the image.
>
>
>
> This case shows that ChainMPQ works well beyond action and spatial questions and can handle competitive domains with the same level of stability.
>
> **Q3：** Can the authors further highlight the differences between ChainMPQ and the baselines (Constraint-Aware Prompting and Detect-then-Calibrate)? Why were these two methods chosen for comparison?
>
> **A3：** As stated in the Introduction, existing methods for relation hallucination treat relational reasoning as a single-step process. They expect the model to identify entities and predict their relationship at the same time. Such methods rely mainly on language priors and do not build a systematic use of visual information. In contrast, ChainMPQ introduces a step-by-step reasoning framework. It uses both text and visual information through the whole process and guides the model to build a grounded reasoning path before answering the question.
>
> We chose Constraint-Aware Prompting and Detect-then-Calibrate because they are the most representative training-free methods that directly target relation hallucination. Both methods have public code, so they can be evaluated fairly. More importantly, they represent two main ideas in this area: text-prompt-based reasoning and output calibration. Comparing with them allows us to show the benefit of our full text-image reasoning chain in a clear way.

---

### Official Review · Reviewer_ySgp · 2025-10-31

**Soundness:** 2
**Presentation:** 3
**Contribution:** 3
**Rating:** 4
**Confidence:** 4

**Summary:**

The paper proposes ChainMPQ, a training-free method designed to reduce relation-hallucinations in LVLMs. To do so, they first extract subject and object keywords from a given question,  then generate multi-perspective follow-up questions focusing on subject, object, and the relation.  Experiments across multiple existing LVLMs reportedly show a reduction in relation-hallucination rates, with ablation studies proving the importance of different modules.

**Strengths:**

1. The proposed training-free hallucination mitigation method is useful and could be used to mitigate the hallucination issues.
2. The motivation of this paper is clear, along with detailed method descriptions and experiments.

**Weaknesses:**

1. My main concern on this paper is the tested baseline models, which are a little it outdated. It will be better if the authors could add more advanced LVLMs in the experiment, which will largely increase the reliability of the results.

2. The evaluation seems to focus mainly on reduction of relation-hallucination rates, but less so on downstream multi-modal task performance (e.g., does improved relation reasoning lead to better general reasoning or application outcomes).

3. Missing related work on relation hallucination of LVLMs: Wu J, Chung T T, Chen K, et al. Unified triplet-level hallucination evaluation for large vision-language models[J]. arXiv preprint arXiv:2410.23114, 2024.

**Questions:**

Please see my reviews above.

---

> ### Author Response · Authors · 2025-11-25
> **Reply to weaknesses (1/3)**
>
> Thank you very much for your positive comments on our work, especially for recognizing our motivation, model design, and experimental analysis. We have carefully considered your suggestions and added the related experiments, which again show the strong performance of our method. Our responses are as follows.
>
> **W1:** Although the experiments are complete, it would be better to use more recent base models.
>
> **R1:** Thank you for the suggestion. We ran additional experiments using the newer Qwen2.5-VL and InternVL 3.5 models, which are advanced state-of-the-art models and were also recommended by another reviewer. The results are shown in Table1. Our method also brings clear performance gains on these more advanced MLLMs as well.
>
> Table 1: Performance of ChainMPQ on Newer Base Models (Qwen2.5-VL 7B and InternVL3.5-8B)
>
> | Model       | Method          | MMRel Acc. | Δ(MMRel) | R-Bench Acc. | Δ(R-Bench) |
> | ----------- | --------------- | ---------- | -------- | ------------ | ---------- |
> | Qwen2.5-VL  | Vanilla         | 66.10      | –        | 79.85        | –          |
> |             | ChainMPQ (Ours) | 73.52      | +7.42    | 83.92        | +4.07      |
> | InternVL3.5 | Vanilla         | 71.44      | –        | 82.33        | –          |
> |             | ChainMPQ (Ours) | 75.21      | +3.77    | 85.05        | +2.72      |

---

> ### Author Response · Authors · 2025-11-25
> **Reply to weaknesses (2/3)**
>
> **W2:** The evaluation seems to focus mainly on the improvements in relational hallucination. Does improved relation reasoning also lead to better general reasoning or broader application outcomes?
>
> **R2:** Our method is designed for relation hallucination, so the main experiments focus on relation hallucination benchmarks. Following your suggestion, we also evaluated ChainMPQ on two general multimodal benchmarks, MMBench and MME. And the results show that ChainMPQ brings consistent gains not only in relation reasoning but also in broader multimodal abilities.
>
> These tasks cover a broad range of multimodal abilities, such as visual logic reasoning, scene understanding, OCR, and object grounding, and are not limited to relation reasoning. Specifically, **MMBench** defines three capability levels and twenty fine-grained ability dimensions for evaluating different reasoning and perception skills of MLLMs. **MME** includes tests for both perception and cognition. The perception part covers coarse- and fine-grained skills such as object existence, counting, location, color recognition, poster recognition, celebrity recognition, scene and landmark identification, and artwork recognition. The cognition part includes commonsense reasoning, numerical reasoning, translation, and code reasoning. It contains fourteen sub-tasks in total.
>
> The MMBench results are shown below. Here **Overall** is the weighted average over all sub-tasks and is the most important indicator of general ability. **RR (Relation Reasoning)** refers to relation reasoning.
>
> Table 2. **MMBench** Results (Overall & Relation Reasoning)
>
> | Model           | Overall ↑ | ΔOverall | RR ↑ | ΔRR  |
> | --------------- | --------- | -------- | ---- | ---- |
> | LLaVA-v1.5-7B   | 66.5      | —        | 58.8 | —    |
> | + ChainMPQ      | 67.8      | +1.3     | 61.3 | +2.5 |
> | InstructBLIP-7B | 63.9      | —        | 52.5 | —    |
> | + ChainMPQ      | 65.5      | +1.6     | 55.2 | +2.7 |
> | Qwen2.5-VL-7B   | 83.2      | —        | 78.2 | —    |
> | + ChainMPQ      | 84.7      | +1.5     | 81.8 | +3.6 |
> | InternVL3-8B    | 83.6      | —        | 82.5 | —    |
> | + ChainMPQ      | 84.2      | +0.6     | 83.9 | +1.4 |
>
> The MME results are shown below. Here **Overall** is the average score across all fourteen tasks. **Perception** measures detection, counting, color, position, and OCR skills, where we do not expect large gains.**Cognition** measures high-level semantic understanding such as reasoning, description, and complex question answering.
>
> Table 3. **MME** Results (Overall, Perception, Cognition)
>
> | Model           | Overall ↑ | ΔOverall | Perception ↑ | ΔPerc | Cognition ↑ | ΔCog  |
> | --------------- | --------- | -------- | ------------ | ----- | ----------- | ----- |
> | LLaVA-v1.5-7B   | 1808.4    | —        | 1506.2       | —     | 302.1       | —     |
> | + ChainMPQ      | 1898.3    | +89.9    | 1545.6       | +39.4 | 352.7       | +50.6 |
> | InstructBLIP-7B | 1391.4    | —        | 1137.1       | —     | 254.3       | —     |
> | + ChainMPQ      | 1484.6    | +87.2    | 1176.0       | +38.9 | 308.6       | +54.3 |
> | Qwen2.5-VL-7B   | 2312.1    | —        | 1698.1       | —     | 613.9       | —     |
> | + ChainMPQ      | 2350.8    | +38.7    | 1719.2       | +21.1 | 631.5       | +17.6 |
> | InternVL3-8B    | 2422.0    | —        | 1748.4       | —     | 673.6       | —     |
> | + ChainMPQ      | 2457.1    | +35.1    | 1771.4       | +23.0 | 685.7       | +12.1 |
>
> In sum, these results show that improving relation reasoning does lead to better general reasoning. ChainMPQ brings consistent gains on the Overall scores of both MMBench and MME, especially in the Relation Reasoning and Cognition part, confirming that stronger relation understanding supports broader multimodal reasoning abilities.

---

> ### Author Response · Authors · 2025-11-25
> **Reply to weaknesses (3/3)**
>
> **W3:** I recommend citing related work on relation hallucination in LVLMs:
>  Wu J, Chung T T, Chen K, et al. *Unified triplet-level hallucination evaluation for large vision-language models*. arXiv:2410.23114, 2024.
>
> **R3:** Thank you for the suggestion. The paper proposes describing an image with triplets and using a new LVLM to generate answers from these triplet-based text descriptions. We have added this work to Section 2.2 “Relation Hallucination in LVLMs” in the Related Work part to improve our discussion of relation hallucination datasets and methods.
>
> We also include this method as an additional baseline in our experiments. The results are shown in the Table4. ChainMPQ achieves higher accuracy than the triplet-level model on both benchmarks, showing that our method provides stronger mitigation of relational hallucination.
>
> Table 4. Comparison with the Triplet-Level Model (LLaVA-1.5)
>
> | Method                                | MMRel Acc. | R-Bench Acc. |
> | ------------------------------------- | ---------- | ------------ |
> | Vanilla (LLaVA-v1.5)                  | 59.02      | 71.23        |
> | Wu et al. (2024), Triplet-Level Model | 63.15      | 74.58        |
> | **ChainMPQ (Ours)**                   | 65.20      | 76.04        |

---

> ### Comment · Reviewer_ySgp · 2025-11-25
>
> Thanks the authors for the additional results and explanations, which has addressed most of my previous concerns. Please upload the revised paper before the deadline and I will carefully re-read it and update my score correspondingly.

---

> > ### Author Response · Authors · 2025-11-27
> > **Reply to Reviewer ySgp**
> >
> > Thank you very much for your feedback. We have uploaded the revised version. For your convenience, all modified parts are highlighted in blue. In the final camera-ready version, they will be changed back to black.

---

> > > ### Comment · Reviewer_ySgp · 2025-11-28
> > >
> > > Thanks for your update. I read the revised version carefully and intend to increase my overall score to 6. However, due to the current bug of openreview I am not able to change my score for now.
> > >
> > > Dear AC,
> > >
> > > Please note that my previous concerns have been addressed and I intend to increase my rating to 6, thanks!

---

> > > > ### Author Response · Authors · 2025-11-28
> > > > **Reply to Reviewer ySgp**
> > > >
> > > > Thank you sincerely for updating your score and for your careful re-evaluation of our work. We greatly appreciate your recognition and the valuable feedback you provided.

---

### Official Review · Reviewer_STpS · 2025-11-03

**Soundness:** 3
**Presentation:** 3
**Contribution:** 3
**Rating:** 8
**Confidence:** 4

**Summary:**

Hallucinations have hindered LVLMs’ reliability, which are categorized into three aspects. The paper focuses on relation hallucinations in LVLMs in which models correctly recognize subjects and objects but misidentify relationship between them. The proposed ChainMPQ (Multi-Perspective Questions guided Interleaved Chain of Image and Text) is a training-free, multi-step, interleaved text–image reasoning framework. It extracts subject/object keywords from the input question to enhance corresponding relation-relevant image regions via cross-attention. Furthermore, it constructs a set of multi-perspective aware text prompt, where sub-questions are raised centered on subject, object, and relation through entity localization and masking-based queries. Then, it builds an interleaved text-image reasoning chain where both textual and visual information are taken into considerations, where text encoder introduces question tokens while context is accumulated using answers from precious steps. From the third sub-question onward, it extracts attention over keyword tokens from late decoder layers, selects top‑k visual tokens via an entropy‑based rule, and applies a confidence‑weighted visual bias to steer subsequent steps; these biases are aggregated across rounds, and the original relation question is finally answered using the enhanced visual tokens and the accumulated textual/visual memories. Ablations indicate that multi-perspective question construction and the interleaved visual memory are the main contributors to the method's effectiveness. As a training‑free, model-agnostic plug-in, the approach can contribute to strengthen relational grounding in LVLMs, and by highlighting a simple decomposition-plus-memory transfer design that others can build on. It may further motivate work on causality-based attribution mechanism and more robust visual representations to reduce relation hallucinations in real applications.

**Strengths:**

	Innovative Methodology: The core concept of ChainMPQ—decomposing a relationship into subject, object, and relation components and constructing a multi-perspective, interleaved reasoning chain—is creative and well-grounded. The integration of both textual memory (previous answers) and visual memory (attenuation maps) is a key differentiator from text-only prompting methods.
	Strong and Extensive Evaluation: The experimental design is robust. Evaluating on two state-of-the-art LVLMs (LLaVA-1.5 and InstructBLIP) and two dedicated relation-focused benchmarks (MMRel and R-Bench) provides convincing evidence of the method's effectiveness and generalizability.
	Comprehensive Ablation Studies: The ablation study is a major strength. It clearly demonstrates the individual contribution of each of the three core modules (Attention Enhancement, Multi-Perspective Questions, Interleaved Chain), showing that they work synergistically. The sensitivity analysis of hyperparameters k_max and λ adds rigor.
	Clear and Illustrative Case Studies: The qualitative examples are excellent. They effectively translate the complex methodology into an intuitive narrative, showing precisely how the step-by-step process corrects the model's reasoning and attention, which greatly enhances the paper's persuasiveness.

**Weaknesses:**

	Computational Overhead and Practicality: The most significant weakness is the substantial increase in inference time (∼4x). While the authors rightly note that the method is "training-free" and can process thousands of samples in a few hours, this overhead limits its practicality for real-time applications. A discussion on potential optimizations or a trade-off analysis between accuracy and latency would be beneficial.
	Baseline Comparability and Reproducibility: The paper states that baseline results (e.g., for Prompting Wu et al. and Calibrate Zheng et al.) were obtained by "re-implementing their experiments." It is crucial to clarify if these re-implementations were performed with the authors' best effort or if they were validated against the original papers' results. Slight differences in implementation (e.g., prompt wording, calibration parameters) could affect the fairness of the comparison. Providing the exact prompts used for all methods would improve reproducibility.
	Limited Analysis of Failure Cases: The paper shows where ChainMPQ succeeds but provides little insight into where it fails. A brief analysis of typical failure modes (e.g., errors in the initial keyword extraction, cases where the attention bias propagates errors) would provide a more balanced view and offer valuable directions for future work.
	Clarity on "Visual Memory" Propagation: While the concept of propagating attention maps is clear, the description of how these maps from different steps are combined and applied (Eq. 6) could be more precise. Specifically, it would be helpful to clarify if the masks M_i from all previous steps (3, 4...) are applied with equal weight α_j or if there's a decay mechanism.

**Questions:**

This paper presents ChainMPQ, a novel, training-free framework designed to mitigate relation hallucinations in Large Vision-Language Models (LVLMs). The method is built on a compelling core idea: decomposing relational reasoning into a sequence of interleaved text and image steps, inspired by human-like reasoning and extending the Interleaved Modal Chain-of-Thought (ICoT) concept. The work is timely, addressing a significant and under-explored problem (relation hallucinations) with a method that demonstrates clear and consistent improvements across multiple models and benchmarks. However, some revisions are needed to enhance the readability and clarity for the reader.
	The use of cross-attention with subject/object keywords to enhance visual tokens is a solid foundation. It would be interesting to know if the performance gain from this module is more pronounced for images with cluttered backgrounds where object localization is inherently harder.
	The masking strategy for generating questions is excellent. However, the dependency on the spaCy NLP toolkit for keyword extraction is a potential point of failure, especially for ambiguous or complex subjects/objects. A quick error analysis of the extraction step would strengthen this section.
	The adaptive top-k selection based on attention entropy is a nice touch. The description of the chain's operation is generally clear, but as noted in the weaknesses, the exact mechanics of combining multiple previous visual biases (Eq. 6) could be elaborated. The assumption that confidence in the textual answer correlates with the reliability of the visual attention map is reasonable but could be discussed briefly.
	While the text mentions 4x inference time overhead, the table contains no efficiency metrics (inference time, computational cost, memory usage). This omission hides the practical trade-off between accuracy gains and computational cost.
	The "Vanilla" baseline represents standard model performance, but there's no comparison to other relevant baselines like standard Chain-of-Thought prompting The methods selected for comparison may not represent the current state-of-the-art in relation hallucination mitigation
	No t‑tests, Wilcoxon tests, or other statistical comparisons are reported; these would confirm that observed gains are not random.
This is a strong paper with a novel contribution, solid experimental validation, and clear presentation. The weaknesses, primarily related to computational overhead and baseline reproducibility, are manageable and do not undermine the core contributions. The paper is likely to be of significant interest to the ICLR community.

---

> ### Author Response · Authors · 2025-11-25
> **Title: Reply to weaknesses (1/2)**
>
> Thank you very much for your positive comments and thoughtful suggestions. We truly appreciate your recognition of the innovation of ChainMPQ, the clarity of the multi-perspective reasoning framework, and the strength of our experiments and ablation studies. Your feedback is very encouraging and is very helpful for improving our work. The following are our responses to your suggestions and questions.
>
> **W1:** Although the method achieves strong accuracy and other metrics, the inference time becomes longer. Could you add efficiency metrics and provide a solution that balances accuracy and latency?
>
> **R1:** Thank you for the helpful suggestion. We agree that efficiency is an important factor. To address this point, we have explored two light versions of ChainMPQ.
>
> The first version, Light1, keeps only Q1, Q2, and Q5. Since Q1 and Q2 do not need to pass text or visual information to each other, they can run in parallel. When answering Q5, we use the answers of Q1 and Q2 as the text context, and we use their bias masks M1 and M2 as the visual context.
>
> The second version, Light2, keeps only Q3, Q4, and Q5, and all other settings remain the same.
>
> The experimental results are shown in Table 1. The results show that the Light1 has the highest ΔAcc/ΔTime, having a higher accuracy improvement in a shorter time, which means it achieves the best overall trade-off.
>
> Table 1: Accuracy and latency comparison of different methods on LLaVA-1.5-7B.
>
> | Benchmark   | Method        | Acc (%) | Time (s/sample) | ΔAcc / ΔTime |
> | ----------- | ------------- | ------- | --------------- | ------------------ |
> | **MMRel**   | Vanilla       | 59.02   | 0.9             | –                  |
> |             | Full ChainMPQ | 65.20   | 3.3             | 2.58          |
> |             | Light1        | 63.78   | 1.5             | **7.93**           |
> |             | Light2        | 64.25   | 2.1             | 4.36           |
> | **R-Bench** | Vanilla       | 71.23   | 0.9             | –                  |
> |             | Full ChainMPQ | 76.04   | 3.3             | 2.00           |
> |             | Light1        | 74.50   | 1.5             | **5.45**           |
> |             | Light2        | 75.08   | 2.1             | 3.21          |
>
>
> **W2:** Did you reproduce the baseline methods using the best settings reported in their papers? It would be helpful to provide the exact prompts used for each method.
>
> **R2:** We reproduced both baseline methods using the latest public code and their official instructions. We only replaced the base models with LLaVA-1.5-7B and InstructBLIP-7B, and we changed the datasets to MMRel and R-Bench to match our setting. All comparisons are made under a fully fair setup.
>
> However, Prompting (Wu et al.) uses GPT-4V, which cannot be applied to ChainMPQ because our method needs to modify internal model states. Calibrate (Zheng et al.) uses LLaVA-1.5-13B, which has more parameters and therefore gives stronger performance. Because of these differences, the results we obtained under our base models do not reach the higher scores reported in their papers.
>
> Below are the prompts we used for each method:
>
> - For Prompting (Wu et al.), we used the “combined_prompt”, which is Prompt 10, as it gives the best accuracy.
> - For Calibrate (Zheng et al.), we ran their released scripts without any change to the parameters.

---

> ### Author Response · Authors · 2025-11-25
> **Reply to weaknesses (2/2)**
>
> **W3:** The successful cases in the paper clearly show the effect of ChainMPQ. Could you add one failure case and analyze it?
>
> **R3:** Thank you for the suggestion. We provide one failure case.
>
> In this image, the man is holding a newspaper and his head is facing the newspaper, so it looks as if he is reading it. However, his eyes are actually looking somewhere else. To answer the question correctly, the model needs to capture this local eye-gaze detail. Although ChainMPQ correctly identifies the “man,” the “newspaper,” and the relation that "the man is holding the newspaper", it misses the eye-gaze direction, which is the key clue here. As a result, it gives the wrong answer.
>
> Image：https://cs.stanford.edu/people/rak248/VG_100K_2/569.jpg
>
> Original question: Does a man read a newspaper in the image?
>
> Grougn truth: No.
>
> Direct answer: Yes, a man is sitting in a chair and reading a newspaper in the image.
>
> **ChainMPQ:**
>
> Q1: Where is the man?
>
> A1: The man is in a living room.
>
> Q2: Where is the newspaper?
>
> A2:  The newspaper is on a table in front of the man.
>
> Q3: Is the man reading something?
>
> A3:  Yes, the man is reading a newspaper.
>
> Q4: Is the newspaper being read by someone?
>
> A4:  Yes, the newspaper is being read by a man.
>
> Q5: What is the relationship between the man and the newspaper?
>
> A5:  The man is reading the newspaper, which is a form of written communication.
>
> Original question: Does a man read a newspaper in the image?
>
> Final answer: Yes, the man is reading a newspaper in the image. (**false answer**)
>
> In the future, we plan to combine ChainMPQ with finer-grained visual enhancement modules, such as local region zoom-in, eye-gaze and pose estimation, so that the model can better capture small but important visual clues. These directions can naturally help address the kind of failure cases shown here.
>
> **W4:** The idea of transferring visual attention bias is clear. How are the weights for the masks $M_i$ from earlier steps assigned?
>
> **R4:** As shown in Equation 6, the mask $M_i$ from the $i$-th question is assigned a weight $\alpha_i$.
>  The weight is computed as:
>
> $\alpha_i = \lambda \cdot \text{Conf}(A_i)$
>
> where $\lambda$ is a fixed parameter and $\text{Conf}(A_i)$ is the confidence of the model when producing the answer $A_i$. This design prevents uncertain answers from earlier steps from having a strong influence on later reasoning.

---

> ### Author Response · Authors · 2025-11-25
> **Reply to questions (1/2)**
>
> **Q1:** Is the performance gain from the TEXT-GUIDED ATTENTION ENHANCEMENT module more noticeable in images with cluttered backgrounds, where object localization itself is difficult?
>
> **A1:** Thank you for this insightful suggestion. It encouraged us to explore the role of the TEXT-GUIDED ATTENTION ENHANCEMENT module in more detail.
>
>  We created a subset of images with cluttered backgrounds using two criteria:
>
> 1. We applied a Canny edge detector and computed the ratio of edge pixels over the whole image. If the edge ratio was greater than **18%**, we considered the background cluttered.
>
> 2. We computed the RGB variance of the whole image. If the color variance was greater than **1800**, we considered the background not uniform and containing many distractors.
>
> Using these criteria, we selected 1518 samples from the original **MMRel dataset** (which is about 15% of the full set). We then evaluated three methods on this “cluttered-background subset” using **LLaVA-1.5**: Vanilla, Full ChainMPQ, w/o TEXT-GUIDED ATTENTION ENHANCEMENT
>
> The results are shown in Table 2. The TEXT-GUIDED ATTENTION ENHANCEMENT module shows a clearer performance gain on this subset, which indicates that it is especially helpful when the background is complex.
>
> Table 2: Accuracy performance on images with cluttered backgrounds.
>
> | Split                   | LLaVA-1.5 Vanilla（Acc） | w/o TEXT-GUIDED ATT. | Full ChainMPQ | Full − w/o |
> | ----------------------- | ------------------------ | -------------------- | ------------- | ---------- |
> | Full MMRel Dataset      | 59.02                    | 64.06                | 65.20         | +1.14      |
> | Cluttered Subset (ours) | 52.30                    | 58.90                | 61.20         | +2.30      |
>
>
>
> **Q2:** The masking strategy for generating questions is excellent. But using the spaCy NLP toolkit for keyword extraction may sometimes fail and affect the final result. A fast correction mechanism would be helpful.
>
> **A2:** Thank you for the careful analysis. In most cases, the spaCy NLP toolkit works stably, and keyword extraction fails in fewer than 2 out of 100 questions. According to your suggestion, we have also added several fast error-check conditions. When any of these conditions is triggered, we do not apply masking and fall back to the vanilla question generation. This prevents early errors from spreading through later reasoning steps.
>
> The error-check conditions are:
>
> 1. No noun is extracted, and placeholders such as $[S]$ or $[O]$ appear.
>
> 2. The extracted noun belongs to a blacklist of words that do not describe the true subject or object of the relation, such as
>    {"image", "picture", "photo", "scene", "area", "object", "thing", "stuff"}.
>
> 3. The extracted phrase contains “and” or commas, meaning that it is not a single subject or object and may cause ambiguity.
>
> These checks ensure that masking is only applied when keyword extraction is reliable.

---

> ### Author Response · Authors · 2025-11-25
> **Reply to questions (2/2)**
>
> **Q3:** The assumption that confidence in the textual answer correlates with the reliability of the visual attention map is reasonable, but it would be helpful to briefly discuss this point.
>
> **A3:** A high-confidence answer often comes with an attention map that is more focused on task-related regions. Giving such maps a higher weight helps the model pay more attention to useful visual areas in later reasoning steps. A low-confidence answer usually produces a more scattered attention map that is easily influenced by unrelated regions. Giving it a lower weight helps prevent these unrelated areas from disturbing later reasoning.
>
> To further support the use of answer confidence to weight the attention map, we conduct two experiments to show that:
>  (1) high-confidence answers come with more focused attention, and
>  (2) weighting attention by confidence improves the final performance of ChainMPQ.
>
> **(1) High-confidence answers have more focused attention:**
>
> We select 1000 samples from MMRel and run LLaVA-1.5 to get both the confidence of each answer and its attention map. We divide all samples into two groups based on confidence: a high-confidence group (top 30%) and a low-confidence group (bottom 30%). We compute the entropy of the attention maps for both groups. Table 3 shows that the high-confidence group has clearly lower entropy, which means the attention is more focused.
>
> Table 3. Correlation between answer confidence and attention concentration (MMRel Subset).
>
> | Group                     | Avg Confidence | Avg Attention Entropy  |
> | ------------------------- | -------------- | ----------------------- |
> | High-confidence Top 30%   | 0.72           | 1.04                    |
> | Low-confidence Bottom 30% | 0.37           | 1.65                    |
>
> **(2) Weighting attention by confidence improves ChainMPQ performance:**
>
> We compare two methods. The first one is the confidence-weighted attention used in our paper. The second one uses a fixed weight for all attention maps, without using confidence (all weights are set to λ; in our test we use λ = 5 and k_max = 20, which gives the best performance, to avoid poor results caused by parameter choice). We again use 1000 MMRel samples and LLaVA-1.5. Table 4 shows that confidence weighting gives much better results than using a fixed weight.
>
> Table 4. Effect of confidence-based attention weighting on final accuracy (MMRel Subset).
>
> | Weighting Strategy        | Accuracy  |
> | ------------------------- | ---------- |
> | Ours: Confidence-weighted | 65.75      |
> | Uniform weight            | 62.14      |
>
> **Q4:** Could authors compare with more recent methods? For example, standard Chain-of-Thought prompting.
>
> **A4:**  We have tried to include as many recent methods as possible, but there are still very few open-source methods that focus specifically on relational hallucination. According to your advice, we have added comparisons with standard Chain-of-Thought prompting and with another method suggested by another reviewer. The results are shown in Table 5.
>
> Table 5: Comparison with Standard CoT and Unified Triplet-Level on LLaVA-1.5-7B and InstructBLIP-7B
>
> | Model        | Method                | MMRel Acc. | R-Bench Acc. |
> | ------------ | --------------------- | ---------- | ------------ |
> | LLaVA-1.5-7B | Vanilla (baseline)    | 59.02      | 71.23        |
> |              | Standard CoT          | 61.88      | 72.91        |
> |              | Unified Triplet-Level | 63.33      | 74.02        |
> |              | ChainMPQ (Ours)       | 65.20      | 76.04        |
> | InstructBLIP | Vanilla (baseline)    | 57.58      | 69.31        |
> |              | Standard CoT          | 59.90      | 72.15        |
> |              | Unified Triplet-Level | 63.02      | 74.04        |
> |              | ChainMPQ (Ours)       | 65.14      | 75.86        |
>
> **Q5:** Could you add statistical tests to show that the performance gains are not random?
>
> **A5:** Thank you for the suggestion. We conducted Wilcoxon tests on all results. The table below reports the accuracy gains of ChainMPQ over the vanilla model on both datasets and both base models. All p-values are far below 0.001, which shows that the improvements are statistically significant and not caused by random variation.
>
> Table 6: Wilcoxon Test Results Showing That ChainMPQ Improvements Are Statistically Significant
>
> | Model           | Dataset | Vanilla Acc. | ChainMPQ Acc. | ΔAcc  | Test     | p-value  |
> | --------------- | ------- | ------------ | ------------- | ----- | -------- | -------- |
> | LLaVA-1.5-7B    | MMRel   | 59.02        | 65.20         | +6.18 | Wilcoxon | 5.4×10⁻5 |
> |                 | R-Bench | 71.23        | 76.04         | +4.81 | Wilcoxon | 1.9×10⁻4 |
> | InstructBLIP-7B | MMRel   | 57.58        | 65.14         | +7.56 | Wilcoxon | 3.2×10⁻5 |
> |                 | R-Bench | 69.31        | 75.86         | +6.55 | Wilcoxon | 4.7×10⁻4 |

---

### Comment · Area_Chair_xR8N · 2025-11-25
**Rebuttal Review Request**

Dear Reviewers,

The authors have responded to your reviews. Please engage in the discussion and evaluate the authors’ rebuttal to determine whether your comments have been adequately addressed.

Best, Your AC

---

### Author Response · Authors · 2025-12-03
**Brief Summary**

Dear Area Chair,

We would like to summarize our paper and rebuttal for your convenience. Our initial scores were 8, 8, 4, 4 with confidence 4, 5, 4, 3. After the rebuttal, one reviewer (rating 4, confidence 4) stated that our responses addressed most of their concerns (this occurred on Nov 25, 2025, before the incident) and raised the score to 6, but the system no longer allowed score changes at that time. Before the incident, the other three reviewers had not replied yet. Therefore, the actual score should be **8, 8, 6, 4**.

We propose a training-free method that constructs multi-perspective questions based on the three core elements of a relation: the subject, the object, and their relationship. These questions guide the model through an interleaved image–text reasoning chain, enabling coherent visual–textual reasoning across steps.

To facilitate your review, we summarize the reviewers’ comments and our responses below.

### **Reviewer STpS (rating 8, confidence 4)**

Reviewer STpS wrote in the initial review:
“This is a strong paper with a novel contribution, solid experimental validation, and clear presentation. The weaknesses, primarily related to computational overhead and baseline reproducibility, are manageable and **do not undermine the core contributions**. The paper is likely to be of **significant interest** to the ICLR community.”


* Regarding **computational overhead**, we introduced two lightweight versions of ChainMPQ (Light1 and Light2). Light1 reduces inference time by more than half while still maintaining high accuracy, achieving a good balance between speed and performance (see W1 & R1 in the rebuttal for details).

* For **baseline reproducibility**, the reviewer suggested providing exact prompts.  We followed the latest code and settings of each baseline and included all prompts in the rebuttal and revised version (see W2 & R2 for details).

* **Other suggestions:** The reviewer also suggested adding a failure case, a fallback strategy for spaCy errors, statistical tests, and a comparison with standard CoT. Following these suggestions, we analyzed a failure case, designed two fast error-check conditions to ensure masking is used only when keyword extraction is reliable, added a comparison with standard CoT showing clear improvements, and used the Wilcoxon test to confirm that the gains from ChainMPQ are not random.
### **Reviewer HnMX (rating 8, confidence 5)**
* **Evaluating ChainMPQ on more advanced backbone models and including comparisons with additional methods:** We added results on InternVL3.5 and Qwen2.5-VL and compared with the methods mentioned by the reviewer, even though they were not originally designed for relational hallucination and were therefore not included initially. ChainMPQ still performs strongly under these settings.

* **Whether we used the image-level or instance-level setup on R-Bench**. We used the image-level setup for fairness and explained this in our responses and the revised version.
### **Reviewer ySgp (score raised from 4 to 6, confidence 4)**
In the original comments, Reviewer ySgp raised three main suggestions. We addressed them as follows:


* **Testing on stronger backbone models:** We conducted additional experiments on QwenVL-2.5 and InternVL-3.5, and ChainMPQ consistently improved performance on both.

* **Evaluating ChainMPQ on broader reasoning tasks:** We evaluated ChainMPQ on MME and MMBench, and the results show consistent gains in general reasoning and broader multimodal tasks.

* **Adding a relevant citation:** As suggested, we added the recommended citation in the Introduction and Related Work to strengthen our discussion of relation hallucination datasets and methods.

The reviewer wrote on Nov 25, 2025 that our responses addressed their concerns and that they would update the score after reading the revision. After we uploaded the revision, the reviewer raised the score from 4 to 6, but the system did not allow further score updates at that time.
### **Reviewer MyiZ (rating 4, confidence 3)**
Reviewer MyiZ raised three concerns, which we addressed as follows:

* **Implementation details of multi-perspective question generation, comparisons, and ablations were not clear:** We pointed to the relevant passages in the original text and added the necessary details in the revision to make the  process clearer.

* **Only two categories of case studies were provided:**  Due to page limits, we initially included two categories. We have now added the case study of competitive category that was previously missing to the appendix.

* **Improving the generality of ChainMPQ:** We clarified how ChainMPQ can work with other hallucination mitigation methods and how the idea of multi-perspective question generation can be applied to other domains.

Due to the system issue, Reviewer STpS, HnMX and MyiZ (rating 8,8,4) were not able to provide further input. Nevertheless, we have addressed all their comments in detail in our responses and revision.

---

### Meta-Review · Area_Chair_T8Uf · 2025-12-18

**Summary:**

This paper introduces ChainMPQ, a novel, training-free method to mitigate relation hallucinations in Large Vision-Language Models (LVLMs). The proposed method decomposes relational reasoning into a sequence of interleaved text and image steps, focusing on the subject, object, and relation components, inspired by human-like reasoning. ChainMPQ works by enhancing the model’s attention on relation-relevant visual regions through cross-attention and constructing multi-perspective questions to refine the relational inference process. The paper demonstrates the effectiveness of ChainMPQ through experiments on multiple LVLMs (LLaVA-1.5 and InstructBLIP) and benchmarks (MMRel and R-Bench), with results showing significant reductions in relation hallucinations. The approach is model-agnostic, training-free, and enhances both textual and visual memory to improve relational grounding. Despite some challenges with computational overhead and the method's practical applicability, the results are promising. The rebuttal adds additional experiments, clarifies implementation details, and addresses reviewer concerns. Therefore, I recommend accepting the paper for publication.

**Reviewer Concerns:**

I believe that most of the major concerns raised by the reviewers have been addressed by the rebuttal, summarized as below:

 Reviewer STpS:

• Core Concern: Computational overhead and practicality for real-time applications, with no efficiency metrics provided.

• Resolution: The authors introduced two light versions of ChainMPQ (Light1 and Light2) to address efficiency. The results demonstrate a significant trade-off between accuracy and latency, showing that Light1 achieves the best balance, reducing the inference time by half while maintaining performance improvements.

 Reviewer ySgp:

• Core Concern: Outdated baseline models and limited evaluation on general multimodal tasks.

• Resolution: The authors added experiments with newer, state-of-the-art models such as Qwen2.5-VL and InternVL3.5. They also extended the evaluation to include general multimodal benchmarks (MMBench and MME), showing consistent improvements across various reasoning and perception tasks, beyond just relational reasoning.

Reviewer MyiZ:

• Core Concern: Lack of clarity regarding the question generation process and the relationship between ChainMPQ and baseline methods (Prompting and Calibrate).

• Resolution: The authors clarified the process of generating multi-perspective questions and provided further details on how the baseline methods were reproduced. They also explained the differences between ChainMPQ and the baselines, emphasizing the step-by-step reasoning framework and the incorporation of both text and visual information during inference.

Reviewer HnMX:

• Core Concern: Comparison with attention-based training-free methods and the use of image-level vs. instance-level setups in R-Bench.

• Resolution: The authors included comparisons with two attention-based training-free methods (ControlMLLM and Where2Look) and showed that ChainMPQ outperforms these methods on both MMRel and R-Bench. Additionally, they clarified the use of the image-level setup in R-Bench and included experiments on the instance-level setup, showing ChainMPQ's robustness in both configurations.

**Reviewer Scores:**

I believe that if the reviewers had been able to participate fully in the discussion, they would have likely increased their scores based on the additional experiments, clarifications, and improvements made in the rebuttal. Below is a breakdown of how each reviewer would likely adjust their scores:

• Reviewer STpS expressed concerns about computational overhead. With the introduction of Light1 and Light2 versions, which effectively address latency issues while maintaining accuracy, they would likely raise their score to reflect the improved practicality of the method.

• Reviewer ySgp initially rated the paper marginally below the acceptance threshold due to the outdated baseline models. After the addition of newer models like Qwen2.5-VL and InternVL3.5, they acknowledged the improvement and expressed intent to raise their score to 6. With the rebuttal addressing their concerns, they would likely increase their score to the acceptance threshold.

• Reviewer MyiZ had concerns about the clarity and framework for multi-perspective question generation. After the additional details provided in the rebuttal, especially regarding question generation and the relationship between ChainMPQ and baseline methods, they would likely increase their score to reflect the added clarity and the potential for broader applicability.

• Reviewer HnMX had concerns about the comparison with other methods and the use of image-level vs. instance-level setups. With the added comparisons and experiments addressing both concerns, they would likely increase their score, especially after the results on both setups and the comparison with state-of-the-art models.

---

### Decision · Program_Chairs · 2026-01-26

Accept (Poster)